# Provably Efficient Q-Learning
# with Low Switching Cost

**Yu Bai**
Stanford University
yub@stanford.edu

**Tengyang Xie**    **Nan Jiang**
UIUC
{tx10, nanjiang}@illinois.edu

**Yu-Xiang Wang**
UC Santa Barbara
yuxiangw@cs.ucsb.edu

## Abstract

We take initial steps in studying PAC-MDP algorithms with limited adaptivity, that is, algorithms that change its exploration policy as infrequently as possible during regret minimization. This is motivated by the difficulty of running fully adaptive algorithms in real-world applications (such as medical domains), and we propose to quantify adaptivity using the notion of *local switching cost*. Our main contribution, Q-Learning with UCB2 exploration, is a model-free algorithm for $H$-step episodic MDP that achieves sublinear regret whose local switching cost in $K$ episodes is $O(H^3 SA \log K)$, and we provide a lower bound of $\Omega(HSA)$ on the local switching cost for any no-regret algorithm. Our algorithm can be naturally adapted to the concurrent setting [13], which yields nontrivial results that improve upon prior work in certain aspects.

## 1 Introduction

This paper is concerned with reinforcement learning (RL) under *limited adaptivity* or *low switching cost*, a setting in which the agent is allowed to act in the environment for a long period but is constrained to switch its policy for at most $N$ times. A small switching cost $N$ restricts the agent from frequently adjusting its exploration strategy based on feedback from the environment.

There are strong practical motivations for developing RL algorithms under limited adaptivity. The setting of restricted policy switching captures various real-world settings where deploying new policies comes at a cost. For example, in medical applications where actions correspond to treatments, it is often unrealistic to execute fully adaptive RL algorithms – instead one can only run a fixed policy approved by the domain experts to collect data, and a separate approval process is required every time one would like to switch to a new policy [19, 2, 3]. In personalized recommendation [25], it is computationally impractical to adjust the policy online based on instantaneous data (for instance, think about online video recommendation where there are millions of users generating feedback at every second). A more common practice is to aggregate data in a long period before deploying a new policy. In problems where we run RL for compiler optimization [4] and hardware placements [20], as well as for learning to optimize databases [18], often it is desirable to limit the frequency of changes to the policy since it is costly to recompile the code, to run profiling, to reconfigure an FPGA devices, or to restructure a deployed relational database. The problem is even more prominent in the RL-guided new material discovery as it takes time to fabricate the materials and setup the experiments [24, 21]. In many of these applications, adaptivity turns out to be really the bottleneck.

Understanding limited adaptivity RL is also important from a theoretical perspective. First, algorithms with low adaptivity (a.k.a. "batched" algorithms) that are as effective as their fully sequential counterparts have been established in bandits [23, 12], online learning [8], and optimization [11], and it would be interesting to extend such undertanding into RL. Second, algorithms with few policy switches are naturally easy to parallelize as there is no need for parallel agents to communicate if

they just execute the same policy. Third, limited adaptivity is closed related to off-policy RL[1] and offers a relaxation less challenging than the pure off-policy setting. We would also like to note that limited adaptivity can be viewed as a constraint for designing RL algorithms, which is conceptually similar to those in constrained MDPs [9, 26].

In this paper, we take initial steps towards studying theoretical aspects of limited adaptivity RL through designing *low-regret algorithms* with limited adaptivity. We focus on model-free algorithms, in particular Q-Learning, which was recently shown to achieve a $\widetilde{O}(\sqrt{\text{poly}(H) \cdot SAT})$ regret bound with UCB exploration and a careful stepsize choice by Jin et al. [16]. Our goal is to design Q-Learning type algorithms that achieve similar regret bounds with a bounded switching cost.

The main contributions of this paper are summarized as follows:

- We propose a notion of *local switching cost* that captures the adaptivity of an RL algorithm in episodic MDPs (Section 2). Algorithms with lower local switching cost will make fewer switches in its deployed policies.

- Building on insights from the UCB2 algorithm in multi-armed bandits [5] (Section 3), we propose our main algorithms, *Q-Learning with UCB2-{Hoeffding, Bernstein} exploration*. We prove that these two algorithms achieve $\widetilde{O}(\sqrt{H^{\{4,3\}}SAT})$ regret (respectively) and $O(H^3 SA \log(K/A))$ local switching cost (Section 4). The regret matches their vanilla counterparts of [16] but the switching cost is only logarithmic in the number of episodes.

- We show how our low switching cost algorithms can be applied in the *concurrent RL* setting [13], in which multiple agents can act in parallel (Section 5). The parallelized versions of our algorithms with UCB2 exploration give rise to *Concurrent Q-Learning* algorithms, which achieve a nearly linear speedup in execution time and compares favorably against existing concurrent algorithms in sample complexity for exploration.

- We show a simple $\Omega(HSA)$ lower bound on the switching cost for any sublinear regret algorithm, which has at most a $O(H^2 \log(K/A))$ gap from the upper bound (Section 7).

## 1.1 Prior work

**Low-regret RL** Sample-efficient RL has been studied extensively since the classical work of Kearns and Singh [17] and Brafman and Tennenholtz [7], with a focus on obtaining a near-optimal policy in polynomial time, i.e. PAC guarantees. A subsequent line of work initiate the study of regret in RL and provide algorithms that achieve regret $\widetilde{O}(\sqrt{\text{poly}(H, S, A) \cdot T})$ [15, 22, 1]. In our episodic MDP setting, the information-theoretic lower bound for the regret is $\Omega(\sqrt{H^2 SAT})$, which is matched in recent work by the UCBVI [6] and ORLC [10] algorithms. On the other hand, while all the above low-regret algorithms are essentially model-based, the recent work of [16] shows that model-free algorithms such as Q-learning are able to achieve $\widetilde{O}(\sqrt{H^{\{4,3\}}SAT})$ regret which is only $O(\sqrt{H})$ worse than the lower bound.

**Low switching cost / batched algorithms** Auer et al. [5] propose UCB2 in bandit problems, which achieves the same regret bound as UCB but has switching cost only $O(\log T)$ instead of the naive $O(T)$. Cesa-Bianchi et al. [8] study the switching cost in online learning in both the adversarial and stochastic setting, and design an algorithm for stochastic bandits that acheive optimal regert and $O(\log \log T)$ switching cost.

Learning algorithms with switching cost bounded by a fixed $O(1)$ constant is often referred to as *batched algorithms*. Minimax rates for batched algorithms have been established in various problems such as bandits [23, 12] and convex optimization [11]. In all these scenarios, minimax optimal $M$-batch algorithms are obtained for all $M$, and their rate matches that of fully adaptive algorithms once $M = O(\log \log T)$.

## 2 Problem setup

In this paper, we consider undiscounted episodic tabular MDPs of the form $(H, \mathcal{S}, \mathbb{P}, \mathcal{A}, r)$. The MDP has horizon $H$ with trajectories of the form $(x_1, a_1, \ldots, x_H, a_H, x_{H+1})$, where $x_h \in \mathcal{S}$ and $a_h \in \mathcal{A}$. The state space $\mathcal{S}$ and action space $\mathcal{A}$ are discrete with $|\mathcal{S}| = S$ and $|\mathcal{A}| = A$. The initial state $x_1$ can be either adversarial (chosen by an adversary who has access to our algorithm), or stochastic specified by some distribution $\mathbb{P}_0(x_1)$. For any $(h, x_h, a_h) \in [H] \times \mathcal{S} \times \mathcal{A}$, the transition probability is denoted as $\mathbb{P}_h(x_{h+1}|x_h, a_h)$. The reward is denoted as $r_h(x_h, a_h) \in [0, 1]$, which we assume to be deterministic[2]. We assume in addition that $r_{h+1}(x) = 0$ for all $x$, so that the last state $x_{H+1}$ is effectively an (uninformative) absorbing state.

A deterministic policy $\pi$ consists of $H$ sub-policies $\pi^h(\cdot) : \mathcal{S} \to \mathcal{A}$. For any deterministic policy $\pi$, let $V_h^\pi(\cdot) : \mathcal{S} \to \mathbb{R}$ and $Q_h^\pi(\cdot, \cdot) : \mathcal{S} \times \mathcal{A} \to \mathbb{R}$ denote its value function and state-action value function at the $h$-th step respectively. Let $\pi_\star$ denote an optimal policy, and $V_h^\star = V_h^{\pi_\star}$ and $Q_h^\star = Q_h^{\pi_\star}$ denote the optimal $V$ and $Q$ functions for all $h$. As a convenient short hand, we denote $[\mathbb{P}_h V_{h+1}](x, a) := \mathbb{E}_{x' \sim \mathbb{P}(\cdot|x,a)}[V_{h+1}(x')]$ and also use $[\widehat{\mathbb{P}}_h V_{h+1}](x_h, a_h) := V_{h+1}(x_{h+1})$ in the proofs to denote observed transition. Unless otherwise specified, we will focus on deterministic policies in this paper, which will be without loss of generality as there exists at least one deterministic policy $\pi_\star$ that is optimal.

**Regret** We focus on the regret for measuring the performance of RL algorithms. Let $K$ be the number of episodes that the agent can play. (so that total number of steps is $T := KH$.) The regret of an algorithm is defined as

$$\text{Regret}(K) := \sum_{k=1}^{K} \left[ V_1^\star(x_1^k) - V_1^{\pi_k}(x_1^k) \right],$$

where $\pi_k$ is the policy it employs before episode $k$ starts, and $V_1^\star$ is the optimal value function for the entire episode.

**Miscellanous notation** We use standard Big-Oh notations in this paper: $A_n = O(B_n)$ means that there exists an absolute constant $C > 0$ such that $A_n \leq CB_n$ (similarly $A_n = \Omega(B_n)$ for $A_n \geq CB_n$). $A_n = \widetilde{O}(B_n)$ means that $A_n \leq C_n B_n$ where $C_n$ depends at most poly-logarithmically on all the problem parameters.

### 2.1 Measuring adaptivity through local switching cost

To quantify the adaptivity of RL algorithms, we consider the following notion of *local switching cost* for RL algorithms.

**Definition 2.1.** *The local switching cost (henceforth also "switching cost") between any pair of policies $(\pi, \pi')$ is defined as the number of $(h, x)$ pairs on which $\pi$ and $\pi'$ are different:*

$$n_{\text{switch}}(\pi, \pi') := \left| \left\{ (h, x) \in [H] \times \mathcal{S} : \pi^h(x) \neq [\pi']^h(x) \right\} \right|.$$

*For an RL algorithm that employs policies $(\pi_1, \ldots, \pi_K)$, its local switching cost is defined as*

$$N_{\text{switch}} := \sum_{k=1}^{K-1} n_{\text{switch}}(\pi_k, \pi_{k+1}).$$

Note that (1) $N_{\text{switch}}$ is a random variable in general, as $\pi_k$ can depend on the outcome of the MDP; (2) we have the trivial bound $n_{\text{switch}}(\pi, \pi') \leq HS$ for any $(\pi, \pi')$ and $N_{\text{switch}}(\mathcal{A}) \leq HS(K-1)$ for any algorithm $\mathcal{A}$.[3]

**Remark** The local switching cost extends naturally the notion of switching cost in online learning [8] and is suitable in scenarios where the cost of deploying a new policy scales with the portion of $(h, x)$

on which the action $\pi^h(x)$ is changed. A closely related notion of adaptivity is the *global switching cost*, which simply measures how many times the algorithm switches its entire policy:

$$N_{\text{switch}}^{\text{gl}} = \sum_{k=1}^{K-1} \mathbf{1}\left\{\pi_k \neq \pi_{k+1}\right\}.$$

As $\pi_k \neq \pi_{k+1}$ implies $n_{\text{switch}}(\pi_k, \pi_{k+1}) \geq 1$, we have the trivial bound that $N_{\text{switch}}^{\text{gl}} \leq N_{\text{switch}}$. However, the global switching cost can be substantially smaller for algorithms that tend to change the policy "entirely" rather than "locally". In this paper, we focus on bounding $N_{\text{switch}}$, and leave the task of tighter bounds on $N_{\text{switch}}^{\text{gl}}$ as future work.

## 3 UCB2 for multi-armed bandits

To gain intuition about the switching cost, we briefly review the UCB2 algorithm [5] on multi-armed bandit problems, which achieves the same regret bound as the original UCB but has a substantially lower switching cost.

The multi-armed bandit problem can be viewed as an RL problem with $H = 1$, $S = 1$, so that the agent needs only play one action $a \in \mathcal{A}$ and observe the (random) reward $r(a) \in [0, 1]$. The distribution of $r(a)$'s are unknown to the agent, and the goal is to achieve low regret.

The UCB2 algorithm is a variant of the celebrated UCB (Upper Confidence Bound) algorithm for bandits. UCB2 also maintains upper confidence bounds on the true means $\mu_1, \ldots, \mu_A$, but instead plays each arm multiple times rather than just once when it's found to maximize the upper confidence bound. Specifically, when an arm is found to maximize the UCB for the $r$-th time, UCB2 will play it $\tau(r) - \tau(r-1)$ times, where

$$\tau(r) = (1+\eta)^r$$

for $r = 0, 1, 2, \ldots$ and some parameter $\eta \in (0, 1)$ to be determined. [4] The full UCB2 algorithm is presented in Algorithm 1.

---

**Algorithm 1** UCB2 for multi-armed bandits

---

**input** Parameter $\eta \in (0, 1)$.
    **Initialize:** $r_j = 0$ for $j = 1, \ldots, A$. Play each arm once. Set $t \leftarrow 0$ and $T \leftarrow T - A$.
    **while** $t \leq T$ **do**
        Select arm $j$ that maximizes $\overline{r}_j + a_{r_j}$, where $\overline{r}_j$ is the average reward obtained from arm $j$ and $a_r = O(\sqrt{\log T / \tau(r)})$ (with some specific choice.)
        Play arm $j$ exactly $\tau(r_j + 1) - \tau(r_j)$ times.
        Set $t \leftarrow t + \tau(r_j + 1) - \tau(r_j)$ and $r_j \leftarrow r_j + 1$.
    **end while**

---

**Theorem 1** (Auer et al. [5]). *For $T \geq \max_{i:\mu_i < \mu^\star} \frac{1}{2\Delta_i^2}$, the UCB2 algorithm achieves expected regret bound*

$$\mathbb{E}\left[\sum_{t=1}^{T}(\mu^\star - \mu_t)\right] \leq O_\eta\left(\log T \cdot \sum_{i:\mu_i < \mu^\star} \frac{1}{\Delta_i}\right),$$

*where $\Delta_i := \mu^\star - \mu_i$ is the gap between arm $i$ and the optimal arm. Further, the switching cost is at most $O(\frac{A \log(T/A)}{\eta})$.*

The switching cost bound in Theorem 1 comes directly from the fact that $\sum_{i=1}^{A}(1+\eta)^{r_i} \leq T$ implies $\sum_{i=1}^{A} r_i \leq O(A \log(T/A)/\eta)$, by the convexity of $r \mapsto (1+\eta)^r$ and Jensen's inequality. Such an approach can be fairly general, and we will follow it in sequel to develop RL algorithm with low switching cost.

# 4 Q-learning with UCB2 exploration

In this section, we propose our main algorithm, Q-learning with UCB2 exploration, and show that it achieves sublinear regret as well as logarithmic local switching cost.

## 4.1 Algorithm description

**High-level idea** Our algorithm maintains wo sets of optimistic $Q$ estimates: a *running estimate* $\widetilde{Q}$ which is updated after every episode, and a *delayed estimate* $Q$ which is only updated occasionally but used to select the action. In between two updates to $Q$, the policy stays fixed, so the number of policy switches is bounded by the number of updates to $Q$.

To describe our algorithm, let $\tau(r)$ be defined as

$$\tau(r) = \lceil (1+\eta)^r \rceil, \quad r = 1, 2, \ldots$$

and define the *triggering sequence* as

$$\{t_n\}_{n \geq 1} = \{1, 2, \ldots, \tau(r_\star)\} \cup \{\tau(r_\star + 1), \tau(r_\star + 2), \ldots\}, \tag{1}$$

where the parameters $(\eta, r_\star)$ will be inputs to the algorithm. Define for all $t \in \{1, 2, \ldots\}$ the quantities

$$\tau_{\text{last}}(t) := \max\{t_n : t_n \leq t\} \quad \text{and} \quad \alpha_t = \frac{H+1}{H+t}.$$

**Two-stage switching strategy** The triggering sequence (1) defines a *two-stage strategy* for switching policies. Suppose for a given $(h, x_h)$, the algorithm decides to take some particular $a_h$ for the $t$-th time, and has observed $(r_h, x_{h+1})$ and updated the running estimate $\widetilde{Q}_h(x_h, a_h)$ accordingly. Then, whether to also update the policy network $Q$ is decided as

- Stage I: if $t \leq \tau(r_\star)$, then always perform the update $Q_h(x_h, a_h) \leftarrow \widetilde{Q}_h(x_h, a_h)$.
- Stage II: if $t > \tau(r_\star)$, then perform the above update only if $t$ is in the triggering sequence, that is, $t = \tau(r) = \lceil (1+\eta)^r \rceil$ for some $r > r_\star$.

In other words, for any state-action pair, the algorithm performs eager policy update in the beginning $\tau(r_\star)$ visitations, and switches to delayed policy update after that according to UCB2 scheduling.

**Optimistic exploration bonus** We employ either a Hoeffding-type or a Bernstein-type exploration bonus to make sure that our running $Q$ estimates are optimistic. The full algorithm with Hoeffding-style bonus is presented in Algorithm 2.

## 4.2 Regret and switching cost guarantee

We now present our main results.

**Theorem 2** (Q-learning with UCB2H exploration achieves sublinear regret and low switching cost). *Choosing* $\eta = \frac{1}{2H(H+1)}$ *and* $r_\star = \left\lceil \frac{\log(10H^2)}{\log(1+\eta)} \right\rceil$, *with probability at least* $1 - p$, *the regret of Algorithm 2 is bounded by* $\widetilde{O}(\sqrt{H^4 SAT})$. *Further, the local switching cost is bounded as* $N_{\text{switch}} \leq O(H^3 SA \log(K/A))$.

Theorem 2 shows that the total regret of Q-learning with UCB2 exploration is $\widetilde{O}(\sqrt{H^4 SAT})$, the same as UCB version of [16]. In addition, the local switching cost of our algorithm is only $O(H^3 SA \log(K/A))$, which is logarithmic in $K$, whereas the UCB version can have in the worst case the trivial bound $HS(K-1)$. We give a high-level overview of the proof Theorem 2 in Section 6, and defer the full proof to Appendix A.

**Bernstein version** Replacing the Hoeffding bonus with a Bernstein-type bonus, we can achieve $\widetilde{O}(\sqrt{H^3 SAT})$ regret ($\sqrt{H}$ better than UCB2H) and the same switching cost bound.

---

**Algorithm 2** Q-learning with UCB2-Hoeffding (UCB2H) Exploration

---

**input** Parameter $\eta \in (0, 1)$, $r_\star \in \mathbb{Z}_{>0}$, and $c > 0$.
    **Initialize:** $\widetilde{Q}_h(x, a) \leftarrow H$, $Q_h \leftarrow \widetilde{Q}_h$, $N_h(x, a) \leftarrow 0$ for all $(x, a, h) \in \mathcal{S} \times \mathcal{A} \times [H]$.
    **for** episode $k = 1, \ldots, K$ **do**
        Receive $x_1$.
        **for** step $h = 1, \ldots, H$ **do**
            Take action $a_h \leftarrow \arg\max_{a'} Q_h(x_h, a')$, and observe $x_{h+1}$.
            $t = N_h(x_h, a_h) \leftarrow N_h(x_h, a_h) + 1$;
            $b_t = c\sqrt{H^3\ell/t}$ (Hoeffding-type bonus);
            $\widetilde{Q}_h(x_h, a_h) \leftarrow (1 - \alpha_t)\widetilde{Q}_h(x_h, a_h) + \alpha_t[r_h(x_h, a_h) + \widetilde{V}_{h+1}(x_{h+1}) + b_t]$.
            $\widetilde{V}_h(x_h) \leftarrow \min\left\{H, \max_{a' \in \mathcal{A}} \widetilde{Q}_h(x_h, a')\right\}$.
            **if** $t \in \{t_n\}_{n \geq 1}$ (where $t_n$ is defined in (1)) **then**
                (Update policy) $Q_h(x_h, \cdot) \leftarrow \widetilde{Q}_h(x_h, \cdot)$.
            **end if**
        **end for**
    **end for**

---

**Theorem 3** (Q-learning with UCB2B exploration achieves sublinear regret and low switching cost)**.** *Choosing* $\eta = \frac{1}{2H(H+1)}$ *and* $r_\star = \left\lceil \frac{\log(10H^2)}{\log(1+\eta)} \right\rceil$, *with probability at least* $1 - p$, *the regret of Algorithm 1 is bounded by* $\widetilde{O}(\sqrt{H^3SAT})$ *as long as* $T = \widetilde{\Omega}(H^6S^2A^2)$. *Further, the local switching cost is bounded as* $N_{\text{switch}} \leq O(H^3SA\log(K/A))$.

The full algorithm description, as well as the proof of Theorem 3, are deferred to Appendix B.

Compared with Q-learning with UCB [16], Theorem 2 and 3 demonstrate that "vanilla" low-regret RL algorithms such as Q-Learning can be turned into low switching cost versions without any sacrifice on the regret bound.

### 4.3 PAC guarantee

Our low switching cost algorithms can also achieve the PAC learnability guarantee. Specifically, we have the following

**Corollary 4** (PAC bound for Q-Learning with UCB2 exploration)**.** *Suppose (WLOG) that* $x_1$ *is deterministic. For any* $\varepsilon > 0$, *Q-Learning with {UCB2H, UCB2B} exploration can output a (stochastic) policy* $\widehat{\pi}$ *such that with high probability*

$$V_1^\star(x_1) - V_1^{\widehat{\pi}}(x_1) \leq \varepsilon$$

*after* $K = \widetilde{O}(H^{\{5,4\}}SA/\varepsilon^2)$ *episodes.*

The proof of Corollary 4 involves turning the regret bounds in Theorem 2 and 3 to PAC bounds using the online-to-batch conversion, similar as in [16]. The full proof is deferred to Appendix C.

## 5 Application: Concurrent Q-Learning

Our low switching cost Q-Learning can be applied to developing algorithms for *Concurrent RL* [13] – a setting in which multiple RL agents can act in parallel and hopefully accelerate the exploration in wall time.

**Setting** We assume there are $M$ agents / machines, where each machine can interact with a independent copy of the episodic MDP (so that the transitions and rewards on the $M$ MDPs are mutually independent). Within each episode, the $M$ machines must play synchronously and cannot communiate, and can only exchange information after the entire episode has finished. Note that our setting is in a way more stringent than [13], which allows communication after each timestep.

We define a "round" as the duration in which the $M$ machines simultanesouly finish one episode and (optionally) communicate and update their policies. We measure the performance of a concurrent

algorithm in its required number of rounds to find an $\varepsilon$ near-optimal policy. With larger $M$, we expect such number of rounds to be smaller, and the best we can hope for is a *linear speedup* in which the number of rounds scales as $M^{-1}$.

**Concurrent Q-Learning**    Intuitively, any low switching cost algorithm can be made into a concurrent algorithm, as its execution can be parallelized in between two consecutive policy switches. Indeed, we can design concurrent versions of our low switching Q-Learning algorithm and achieve a nearly linear speedup.

**Theorem 5** (Concurrent Q-Learning achieves nearly linear speedup). *There exists concurrent versions of Q-Learning with {UCB2H, UCB2B} exploration such that, given a budget of $M$ parallel machines, returns an $\varepsilon$ near-optimal policy in*

$$\widetilde{O}\left(H^3 SA + \frac{H^{\{5,4\}} SA}{\varepsilon^2 M}\right)$$

*rounds of execution.*

Theorem 5 shows that concurrent Q-Learning has a linear speedup so long as $M = \widetilde{O}(H^{\{2,1\}}/\varepsilon^2)$. In particular, in high-accuracy (small $\varepsilon$) cases, the constant overhead term $H^3 SA$ can be negligible and we essentially have a linear speedup over a wide range of $M$. The proof of Theorem 5 is deferred to Appendix D.

**Comparison with existing concurrent algorithms**    Theorem 5 implies a PAC mistake bound as well: there exists concurrent algorithms on $M$ machines, Concurrent Q-Learning with {UCB2H, UCB2B}, that performs a $\varepsilon$ near-optimal action on all but

$$\widetilde{O}\left(H^4 SAM + \frac{H^{\{6,5\}} SA}{\varepsilon^2}\right) := N_\varepsilon^{\mathsf{CQL}}$$

actions with high probability (detailed argument in Appendix D.2).

We compare ourself with the Concurrent MBIE (CMBIE) algorithm in [13], which considers the discounted and infinite-horizon MDPs, and has a mistake bound[5]

$$\widetilde{O}\left(\frac{S'A'M}{\varepsilon(1-\gamma')^2} + \frac{S'^2 A'}{\varepsilon^3(1-\gamma')^6}\right) := N_\varepsilon^{\mathsf{CMBIE}}$$

Our concurrent Q-Learning compares favorably against CMBIE in terms of the mistake bound:

- Dependence on $\varepsilon$. CMBIE achieves $N_\varepsilon^{\mathsf{CMBIE}} = \widetilde{O}(\varepsilon^{-3} + \varepsilon^{-1} M)$, whereas our algorithm achieves $N_\varepsilon^{\mathsf{CQL}} = \widetilde{O}(\varepsilon^{-2} + M)$, better by a factor of $\varepsilon^{-1}$.
- Dependence on $(H, S, A)$. These are not comparable in general, but under the "typical" correspondence[6] $S' \leftarrow HS$, $A' \leftarrow A$, $(1-\gamma')^{-1} \leftarrow H$, we get $N_\varepsilon^{\mathsf{CMBIE}} = \widetilde{O}(H^3 SAM\varepsilon^{-1} + H^8 S^2 A\varepsilon^{-3})$. Compared to $N_\varepsilon^{\mathsf{CQL}}$, CMBIE has a higher dependence on $H$ as well as a $S^2$ term due to its model-based nature.

## 6    Proof overview of Theorem 2

The proof of Theorem 2 involves two parts: the switching cost bound and the regret bound. The switching cost bound results directly from the UCB2 switching schedule, similar as in the bandit case (cf. Section 3). However, such a switching schedule results in delayed policy updates, which makes establishing the regret bound technically challenging.

The key to the $\widetilde{O}(\mathrm{poly}(H) \cdot \sqrt{SAT})$ regret bound for "vanilla" Q-Learning in [16] is a *propagation of error* argument, which shows that the regret[7] from the $h$-th step and forward (henceforth the

$h$-regret), defined as

$$\sum_{k=1}^{K} \widetilde{\delta}_h^k := \sum_{k=1}^{K} \left[ \widetilde{V}_h^k - V_h^{\pi_k} \right] (x_h^k),$$

is bounded by $1 + 1/H$ times the $(h+1)$-regret, plus some bounded error term. As $(1+1/H)^H = O(1)$, this fact can be applied recursively for $h = H, \dots, 1$ which will result in a total regret bound that is not exponential in $H$. The control of the (excess) error propagation factor by $1/H$ and the ability to converge are then achieved simultaneously via the stepsize choice $\alpha_t = \frac{H+1}{H+t}$.

In constrast, our low-switching version of Q-Learning updates the exploration policy in a delayed fashion according to the UCB2 schedule. Specifically, the policy at episode $k$ does not correspond to the argmax of the running estimate $\widetilde{Q}^k$, but rather a previous version $Q^k = \widetilde{Q}^{k'}$ for some $k' \leq k$. This introduces a mismatch between the $Q$ used for exploration and the $Q$ being updated, and it is a priori possible whether such a mismatch will blow up the propagation of error.

We resolve this issue via a novel error analysis, which at a high level consists of the following steps:

(i) We show that the quantity $\widetilde{\delta}_h^k$ is upper bounded by a *max error*

$$\widetilde{\delta}_h^k \leq \left( \max\left\{ \widetilde{Q}_h^{k'}, \widetilde{Q}_h^k \right\} - Q_h^{\pi_k} \right) (x_h^k, a_h^k) = \left( \widetilde{Q}_h^{k'} - Q_h^{\pi_k} + \left[ \widetilde{Q}_h^k - \widetilde{Q}_h^{k'} \right]_+ \right) (x_h^k, a_h^k)$$

(Lemma A.3). On the right hand side, the first term $\widetilde{Q}_h^{k'} - Q_h^{\pi_k}$ does not have a mismatch (as $\pi_k$ depends on $\widetilde{Q}^{k'}$) and can be bounded similarly as in [16]. The second term $[\widetilde{Q}_h^k - \widetilde{Q}_h^{k'}]_+$ is a perturbation term, which we bound in a precise way that relates to stepsizes in between episodes $k'$ to $k$ and the $(h+1)$-regret (Lemma A.4).

(ii) We show that, under the UCB2 scheduling, the combined error above results a mild blowup in the relation between $h$-regret and $(h+1)$-regret – the multiplicative factor can be now bounded by $(1+1/H)(1+O(\eta H))$ (Lemma A.5). Choosing $\eta = O(1/H^2)$ will make the multiplicative factor $1 + O(1/H)$ and the propagation of error argument go through.

We hope that the above analysis can be applied more broadly in analyzing exploration problems with delayed updates or asynchronous parallelization.

## 7 Lower bound on switching cost

**Theorem 6.** *Let $A \geq 4$ and $\mathcal{M}$ be the set of episodic MDPs satisfying the conditions in Section 2. For any RL algorithm $\mathcal{A}$ satisfying $N_{\text{switch}} \leq HSA/2$, we have*

$$\sup_{M \in \mathcal{M}} \mathbb{E}_{x_1, M} \left[ \sum_{k=1}^{K} V_1^\star(x_1) - V_1^{\pi_k}(x_1) \right] \geq KH/4.$$

*i.e. the worst case regret is linear in $K$.*

Theorem 6 implies that the switching cost of any no-regret algorithm is lower bounded by $\Omega(HSA)$, which is quite intuitive as one would like to play each action at least once on all $(h, x)$. Compared with the lower bound, the switching cost $O(H^3 SA \log K)$ we achieve through UCB2 scheduling is at most off by a factor of $O(H^2 \log K)$. We believe that the $\log K$ factor is not necessary as there exist algorithms achieving double-log [8] in bandits, and would also like to leave the tightening of the $H^2$ factor as future work. The proof of Theorem 6 is deferred to Appendix E.

## 8 Conclusion

In this paper, we take steps toward studying limited adaptivity RL. We propose a notion of local switching cost to account for the adaptivity of RL algorithms. We design a Q-Learning algorithm with infrequent policy switching that achieves $\widetilde{O}(\sqrt{H^{\{4,3\}} SAT})$ regret while switching its policy for at most $O(\log T)$ times. Our algorithm works in the concurrent setting through parallelization and achieves nearly linear speedup and favorable sample complexity. Our proof involves a novel

perturbation analysis for exploration algorithms with delayed updates, which could be of broader interest.

There are many interesting future directions, including (1) low switching cost algorithms with tighter regret bounds, most likely via model-based approaches; (2) algorithms with even lower switching cost; (3) investigate the connection to other settings such as off-policy RL.

## Acknowledgment

The authors would like to thank Emma Brunskill, Ramtin Keramati, Andrea Zanette, and the staff of CS234 at Stanford for the valuable feedback at an earlier version of this work, and Chao Tao for the very insightful feedback and discussions on the concurrent Q-learning algorithm. YW was supported by a start-up grant from UCSB CS department, NSF-OAC 1934641 and a gift from AWS ML Research Award.

## Footnotes

[1]In particular, $N = 0$ corresponds to off-policy RL, where the algorithm can only choose one data collection policy [14].

[2]Our results can be straightforwardly extended to the case with stochastic rewards.

[3]To avoid confusion, we also note that our local switching cost is not to measure the change of the sub-policy $\pi^h$ between timestep $h$ and $h+1$ (which is in any case needed due to potential non-stationarity), but rather to measure the change of the entire policy $\pi_k = \left\{ \pi_k^h \right\}$ between episode $k$ and $k+1$.

[4]For convenience, here we treat $(1+\eta)^r$ as an integer. In Q-learning we could not make this approximation (as we choose $\eta$ super small), and will massage the sequence $\tau(r)$ to deal with it.

[5]$(S', A', \gamma')$ are the {# states, # actions, discount factor} of the discounted infinite-horizon MDP.

[6]One can transform an episodic MDP with $S$ states to an infinite-horizon MDP with $HS$ states. Also note that the "effective" horizon for discounted MDP is $(1-\gamma)^{-1}$.

[7]Technically it is an upper bound on the regret.

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
