[Supplementary Material]

# Supplementary Materials: Provably Efficient Q-Learning with Low Switching Cost

**Yu Bai**
Stanford University
yub@stanford.edu

**Tengyang Xie**   **Nan Jiang**
UIUC
{tx10, nanjiang}@illinois.edu

**Yu-Xiang Wang**
UC Santa Barbara
yuxiangw@cs.ucsb.edu

## A   Proof of Theorem 2

This section is structured as follows. We collect notation in Section A.1 and list some basic properties of the running estimate $\widetilde{Q}$ in Section A.2, establish useful perturbation bounds on $[\widetilde{Q}_h^k - \widetilde{Q}_h^{k'}]_+$ in Section A.3, and present the proof of the main theorem in Section A.4.

### A.1   Notation

Let $\widetilde{Q}_h^k(x, a)$ and $Q_h^k(x, a)$ denote the estimates $\widetilde{Q}$ and $Q$ in Algorithm 2 before the $k$-th episode has started. Note that $\widetilde{Q}_h^1(x, a) = Q_h^1(x, a) \equiv H$.

Define the sequences

$$\alpha_t^0 := \prod_{i=1}^t (1 - \alpha_i), \ \ \alpha_t^i := \alpha_i \cdot \prod_{\tau=i+1}^t (1 - \alpha_\tau).$$

For $t \geq 1$, we have $\alpha_t^0 = 0$ and $\sum_{i=1}^t \alpha_t^i = 1$. For $t = 0$, we have $\alpha_t^0 = 1$.

With the definition of $\alpha_t^i$ in hand, we have the following explicit formula for $\widetilde{Q}_h^k$:

$$\widetilde{Q}_h^k(x, a) = \alpha_t^0 H + \sum_{i=1}^t \alpha_t^i \left( r_h(x, a) + \widetilde{V}_{h+1}^{k_i}(x_{h+1}^{k_i}) + b_i \right),$$

where $t$ is the number of updates on $\widetilde{Q}_h(x, a)$ **prior to** the $k$-th epoch, and $k_1, \ldots, k_t$ are the indices for the epochs. Note that $k = k_{t+1}$ if the algorithm indeed observes $x$ and takes the action $a$ on the $h$-th step of episode $k$.

Throughout the proof we let $\ell := \log(SAT/p)$ denote a log factor, where we recall $p$ is the pre-specified tail probability.

### A.2   Basics

**Lemma A.1** (Properties of $\alpha_t^i$; Lemma 4.1, [1]). *The following properties hold for the sequence $\alpha_t^i$:*

*(a)* $\frac{1}{\sqrt{t}} \leq \sum_{i=1}^t \frac{\alpha_t^i}{\sqrt{i}} \leq \frac{2}{\sqrt{t}}$ *for every $t \geq 1$.*

*(b)* $\max_{i \in [t]} \alpha_t^i \leq \frac{2H}{t}$ *and* $\sum_{i=1}^t (\alpha_t^i)^2 \leq \frac{2H}{t}$ *for every $t \geq 1$.*

*(c)* $\sum_{t=i}^\infty \alpha_t^i = 1 + \frac{1}{H}$ *for every $i \geq 1$.*

**Lemma A.2** ($\widetilde{Q}$ is optimistic and accurate; Lemma 4.2 & 4.3, [1]). *We have for all $(h, x, a, k) \in [H] \times \mathcal{S} \times \mathcal{A} \times [K]$ that*

$$\widetilde{Q}_h^k(x,a) - Q_h^\star(x,a)$$

$$= \alpha_t^0(H - Q_h^\star(x,a)) + \sum_{i=1}^t \alpha_t^i \left( r_h(x,a) + \widetilde{V}_{h+1}^{k_i}(x_{h+1}^{k_i}) - V_{h+1}^\star(x_{h+1}^{k_i}) + \left[ \left( \widehat{\mathbb{P}}_h^{k_i} - \mathbb{P}_h \right) V_{h+1}^\star \right](x,a) + b_i \right),$$

*where* $[\widehat{\mathbb{P}}_h^{k_i} V_{h+1}](x,a) := V_{h+1}(x_{h+1}^{k_i})$.

*Further, with probability at least $1 - p$, choosing $b_t = c\sqrt{H^3 \ell / t}$ for some absolute constant $c > 0$, we have for all $(h, x, a, k)$ that*

$$0 \le \widetilde{Q}_h^k(x,a) - Q_h^\star(x,a) \le \alpha_t^0 H + \sum_{i=1}^t \alpha_t^i (\widetilde{V}_{h+1}^{k_i} - V_{h+1}^\star)(x_{h+1}^{k_i}) + \beta_t$$

*where $\beta_t := 2 \sum_{i=1}^t \alpha_t^i b_i \le 4c\sqrt{H^3 \ell / t}$.*

**Remark.** This first part of the Lemma, i.e. the expression of $\widetilde{Q}_h^k - Q_h^\star$ in terms of rewards and value functions, is an aggregated form for the $Q$ functions under the Q-Learning updates, and is independent to the actual exploration policy as well as the bonus.

### A.3 Perturbation bound under delayed Q updates

For any $(h, k) \in [H] \times [K]$, let

$$\widetilde{\delta}_h^k := \left( \widetilde{V}_h^k - V_h^{\pi_k} \right)(x_h^k), \quad \widetilde{\phi}_h^k := \left( \widetilde{V}_h^k - V_h^\star \right)(x_h^k)$$

denote the errors of the estimated $\widetilde{V}_h^k$ relative to $V^{\pi_k}$ and $V^\star$. As $\widetilde{Q}$ is optimistic, the regret can be bounded as

$$\text{Regret}(K) = \sum_{k=1}^K \left[ V_1^\star(x_1^k) - V_1^{\pi_k}(x_1^k) \right] \le \sum_{k=1}^K \left[ \widetilde{V}_1^k(x_1^k) - V_1^{\pi_k}(x_1^k) \right] = \sum_{k=1}^K \widetilde{\delta}_1^k.$$

The goal of the propagation of error is to related $\sum_{k=1}^K \widetilde{\delta}_h^k$ by $\sum_{k=1}^K \widetilde{\delta}_{h+1}^k$.

We begin by showing that $\widetilde{\delta}_h^k$ is controlled by the max of $\widetilde{Q}_h^k$ and $\widetilde{Q}_h^{k'}$, where $k' = k_{\tau_{\text{last}}(t)+1}$.

**Lemma A.3** (Max error under delayed policy update). *We have*

$$\widetilde{\delta}_h^k \le \left( \max \left\{ \widetilde{Q}_h^{k'}, \widetilde{Q}_h^k \right\} - Q_h^{\pi_k} \right)(x_h^k, a_h^k) = \left( \widetilde{Q}_h^{k'} - Q_h^{\pi_k} + \left[ \widetilde{Q}_h^k - \widetilde{Q}_h^{k'} \right]_+ \right)(x_h^k, a_h^k). \quad (1)$$

*where $k' = k_{\tau_{\text{last}}(t)+1}$ (which depends on k.) In particular, if $t = \tau_{\text{last}}(t)$, then $k = k'$ and the upper bound reduces to $(\widetilde{Q}_h^{k'} - Q_h^{\pi_k})(x_h^k, a_h^k)$.*

*Proof.* We first show (1). By definition of $\pi_k$ we have $V_h^{\pi_k}(x_h^k) = Q_h^{\pi_k}(x_h^k, a_h^k)$,

so it suffices to show that

$$\widetilde{V}_h^k(x_h^k) \le \max \left\{ \widetilde{Q}_h^k(x_h^k, a_h^k), \widetilde{Q}_h^{k'}(x_h^k, a_h^k) \right\}.$$

Indeed, we have

$$\widetilde{V}_h^k(x_h^k) = \min \left\{ H, \max_{a'} \widetilde{Q}_h^k(x_h^k, a') \right\} \le \max_{a'} \widetilde{Q}_h^k(x_h^k, a').$$

On the other hand, $a_h^k$ maximizes $Q_h(x_h^k, \cdot)$. Due to the scheduling of the delayed update, $Q_h(x_h^k, \cdot)$ was set to $\widetilde{Q}_h^{k_{\tau_{\text{last}}(t)+1}}(x_h^k, \cdot)$, and $\widetilde{Q}_h^{\widetilde{k}}(x_h^k, a_h^k)$ was not updated since then before $\widetilde{k} = k' = k_{\tau_{\text{last}}(t)+1}$, so $Q_h(x_h^k, \cdot) = \widetilde{Q}_h^{k'}(x_h^k, \cdot)$.

Now, defining

$$q_{\text{old}}(\cdot) := \widetilde{Q}_h^{k'}(x_h^k, \cdot), \quad q_{\text{new}}(\cdot) := \widetilde{Q}_h^k(x_h^k, \cdot),$$

the vectors $q_{\mathrm{old}}$ and $q_{\mathrm{new}}$ only differ in the $a_h^k$-th component (which is the only action taken therefore also the only component that is updated). If $q_{\mathrm{new}}$ is also maximized at $a_h^k$, then we have $\widetilde{V}_h^k(x_h^k) \leq q_{\mathrm{new}}(a_h^k)$; otherwise it is maximized at some $a' \neq a_h^k$ and we have

$$\widetilde{V}_h^k(x_h^k) \leq q_{\mathrm{new}}(a') = q_{\mathrm{old}}(a') \leq \max_a q_{\mathrm{old}}(a) = \widetilde{Q}_h^{k'}(x_h^k, a_h^k).$$

Putting together we get

$$\widetilde{V}_h^k(x_h^k) \leq \max\left\{\widetilde{Q}_h^k(x_h^k, a_h^k), \widetilde{Q}_h^{k'}(x_h^k, a_h^k)\right\},$$

which implies (1).

$\square$

Lemma A.3 suggests bounding $\widetilde{\delta}_h^k$ via bounding the "main term" $\widetilde{Q}^{k'} - Q_h^{\pi_k}$ and "perturbation term" $[\widetilde{Q}_h^k - \widetilde{Q}_h^{k'}]_+$ separately. We now establish the bound on the perturbation term.

**Lemma A.4** (Perturbation bound on $(\widetilde{Q}_h^k - \widetilde{Q}_h^{k'})_+$). *For any $k$ such that $k > k'$ (so that the perturbation term is non-zero), we have*

$$\left[\widetilde{Q}_h^k - \widetilde{Q}_h^{k'}\right]_+ (x_h^k, a_h^k) \leq \beta_t + \sum_{i=\tau_{\mathrm{last}}(t)+1}^{t} \alpha_t^i \widetilde{\phi}_{h+1}^{k_i} + \overline{\zeta}_h^k,$$

*where*

$$\overline{\zeta}_h^k := \left| \sum_{i=\tau_{\mathrm{last}}(t)+1}^{t} \alpha_t^i [(\widehat{\mathbb{P}}_h^k - \mathbb{P}_h) V_{h+1}^\star](x_h^k, a_h^k) \right|$$

*and w.h.p. we have uniformly over all $(h, k)$ that $\overline{\zeta}_h^k \leq C\sqrt{H^3 \ell / t}$ for some absolute constant $C > 0$.*

*Proof.* Throughout this proof we will omit the arguments $(x_h^k, a_h^k)$ in $\widetilde{Q}_h$ and $r_h$ as they are clear from the context. By the update formula for $\widetilde{Q}$ in Algorithm 2, we get

$$\widetilde{Q}_h^k = \left(\prod_{i=\tau_{\mathrm{last}}(t)+1}^{t} (1 - \alpha_i)\right) \widetilde{Q}_h^{k'} + \sum_{i=\tau_{\mathrm{last}}(t)+1}^{t} \alpha_t^i \left[r_h(x_h^k, a_h^k) + \widetilde{V}_{h+1}^{k_i}(x_{h+1}^{k_i}) + b_i\right].$$

Subtracting $\widetilde{Q}_h^{k'}$ on both sides (and noting that $\left(\prod_{i=\tau_{\mathrm{last}}(t)+1}^{t}(1 - \alpha_i)\right) + \sum_{i=\tau_{\mathrm{last}}(t)+1}^{t} \alpha_t^i = 1$), we get

$$\widetilde{Q}_h^k - \widetilde{Q}_h^{k'} = \sum_{i=\tau_{\mathrm{last}}(t)+1}^{t} \alpha_t^i \underbrace{\left[r_h + \widetilde{V}_{h+1}^{k_i}(x_{h+1}^{k_i}) + b_i - \widetilde{Q}_h^{k'}\right]}_{d_i}. \tag{2}$$

We now upper bound $d_i$ for each $i$. Adding and subtracting $Q_h^\star$, we obtain

$$d_i = \left(r_h + \widetilde{V}_{h+1}^{k_i}(x_{h+1}^{k_i}) + b_i - Q_h^\star\right) - (\widetilde{Q}_h^{k'} - Q_h^\star)$$

$$\overset{(i)}{=} \widetilde{V}_{h+1}^{k_i}(x_{h+1}^{k_i}) - V^\star(x_{h+1}^{k_i}) + (\widehat{\mathbb{P}}_h^k - \mathbb{P}_h)V_{h+1}^\star + b_i - \left(\widetilde{Q}_h^{k'} - Q_h^\star\right)$$

$$\overset{(ii)}{\leq} b_i + \widetilde{\phi}_{h+1}^{k_i} + \underbrace{(\widehat{\mathbb{P}}_h^k - \mathbb{P}_h)V_{h+1}^\star}_{:=\zeta_i}.$$

where (i) follows from the Bellman optimality equation on $Q_h^\star$, and that $[\widehat{\mathbb{P}}_h^k V_{h+1}^\star](x_h^k, a_h^k) = V_{h+1}^\star(x_{h+1}^k)$ and (ii) follows from the optimistic property of $\widetilde{Q}_h^{k'}$ (from Lemma A.2) and the definition of $\widetilde{\phi}_{h+1}^{k_i}$. Substituting this into (2) gives

$$\left[\widetilde{Q}_h^k - \widetilde{Q}_h^{k'}\right]_+ \leq \left[\sum_{i=\tau_{\mathrm{last}}(t)+1}^{t} \alpha_t^i \left(b_i + \widetilde{\phi}_{h+1}^{k_i} + \zeta_i\right)\right]_+ \leq \beta_t + \sum_{i=\tau_{\mathrm{last}}(t)+1}^{t} \alpha_t^i \widetilde{\phi}_{h+1}^{k_i} + \underbrace{\left|\sum_{i=\tau_{\mathrm{last}}(t)+1}^{t} \alpha_t^i \zeta_i\right|}_{\overline{\zeta}_h^k}.$$

Finally, note that $\zeta_i$ is a martingale difference sequence, so we can apply the Azuma-Hoeffding inequality to get that

$$\bar{\zeta}_h^k \le c \sqrt{\sum_{i=\tau_{\text{last}}(t)+1}^{t} (\alpha_t^i)^2 H^2 \ell} \overset{(i)}{\le} c\sqrt{\frac{2H}{t} \cdot H^2 \ell} = C\sqrt{\frac{H^3 \ell}{t}}$$

uniformly over $(h, k)$, where (i) follows from Lemma A.1(b). $\qquad \square$

### A.4    Proof of Theorem 2

Proof of the main theorem is done through combining the perturbation bound and the "main term", and showing that the propagation of error argument still goes through.

**Lemma A.5** (Error accumulation under delayed update). *Suppose we choose* $\eta = \frac{1}{2H(H+1)}$ *and* $r_\star = \left\lceil \frac{\log(10H^2)}{\log(1+\eta)} \right\rceil$ *for the triggering sequence* (1) *then we have for all $i$ that*

$$\sum_{t:t\ge i, \tau_{\text{last}}(t)\le i-1} \alpha_t^i + \sum_{t:\tau_{\text{last}}(t)\ge i} \alpha_{\tau_{\text{last}}(t)}^i \le 1 + 3/H.$$

*Proof.* Let $\widetilde{S}_i$ denote the above sum. We compare $\widetilde{S}_i$ with

$$S_i := \sum_{t=i}^{\infty} \alpha_t^i = 1 + \frac{1}{H},$$

where the last equality follows from Lemma A.1(c).

Let us consider $\widetilde{S}_i - S_i$ by looking at the difference of the individual terms for each $t \ge i$. When taking the difference, the term $\sum_{t:t\ge i, \tau_{\text{last}}(t)\le i-1} \alpha_t^i$ will vanish, and all terms in $\sum_{t:\tau_{\text{last}}(t)\ge i} \alpha_{\tau_{\text{last}}(t)}^i$ will vanish if $\tau_{\text{last}}(t) = t$. By the design of the triggering sequence $\{t_n\}$, we know that this happens for all $t \le \tau(r_\star)$, so we have

$$\widetilde{S}_i - S_i = \sum_{t:\tau_{\text{last}}(t)\ge i; t>\tau(r_\star)} \alpha_{\tau_{\text{last}}(t)}^i - \alpha_t^i.$$

Let $r(i) = \min\{r : \tau(r) \ge i\}$, then the above can be rewritten as

$$\widetilde{S}_i - S_i = \sum_{r\ge \max\{r_\star, r(i)\}} \sum_{t=\tau(r)}^{\tau(r+1)-1} \alpha_{\tau(r)}^i - \alpha_t^i.$$

For each $t$ (and associated $r \ge r_\star$), we have the bound

$$\alpha_{\tau(r)}^i - \alpha_t^i = \alpha_t^i \left[ \prod_{j=\tau(r)+1}^{t} (1-\alpha_j)^{-1} - 1 \right] = \alpha_t^i \left[ \prod_{j=\tau(r)+1}^{t} \left(1 - \frac{H+1}{H+j}\right)^{-1} - 1 \right]$$

$$= \alpha_t^i \left[ \prod_{j=\tau(r)+1}^{t} \left(1 + \frac{H+1}{j-1}\right) - 1 \right] \le \alpha_t^i \left[ \left(1 + \frac{H+1}{\tau(r)}\right)^{t-\tau(r)} - 1 \right]$$

$$\le \alpha_t^i \left[ \left(1 + \frac{H+1}{\tau(r)}\right)^{\tau(r+1)-\tau(r)-1} - 1 \right] \overset{(i)}{\le} \alpha_t^i \left[ \left(1 + \frac{H+1}{\tau(r)}\right)^{\eta\tau(r)} - 1 \right]$$

$$\overset{(ii)}{\le} \alpha_t^i \left[ e^{\eta(H+1)} - 1 \right] \le \alpha_t^i \cdot 2\eta(H+1).$$

In the above, (i) holds as we have

$$\tau(r+1) - 1 - \tau(r) = \lceil (1+\eta)^{r+1} \rceil - 1 - \lceil (1+\eta)^r \rceil \le (1+\eta)^{r+1} - (1+\eta)^r \le \eta\tau(r),$$

and (ii) holds whenever $\eta(H+1) \leq 1/2$. Choosing

$$\eta = \frac{1}{2H(H+1)} \quad \text{and} \quad r_\star = \left\lceil \frac{\log(10H^2)}{\log(1+\eta)} \right\rceil \leq 8H^2 \log(10H^2),$$

the above requirement will be satisfied. Therefore we have

$$\widetilde{S}_i - S_i \leq 2\eta(H+1) \sum_{r \geq \max\{r_\star, r(i)\}} \sum_{t=\tau(r)}^{\tau(r+1)-1} \alpha_t^i \leq 2\eta(H+1) \sum_{t=i}^{\infty} \alpha_t^i = \frac{1}{H} S_i,$$

and thus

$$\widetilde{S}_i \leq \left(1 + \frac{1}{H}\right) S_i \leq 1 + \frac{3}{H}.$$

$\square$

We are now in position to prove the main theorem.

**Theorem 2** (Q-learning with UCB2H, restated). *Choosing $\eta = \frac{1}{2H(H+1)}$ and $r_\star = \left\lceil \frac{\log(10H^2)}{\log(1+\eta)} \right\rceil$, with probability at least $1 - p$, the regret of Algorithm 2 is bounded by $O(\sqrt{H^4SAT\ell})$, where $\ell := \log(SAT/p)$ is a log factor. Further, the local switching cost is bounded as $N_{\text{switch}} \leq O(H^3SA\log(K/A))$.*

**Proof of Theorem 2** The proof consists of two parts: upper bounding the regret, and upper bounding the local switching cost.

**Part I: Regret bound** By Lemma A.3, we have

$$\widetilde{\delta}_h^k \leq \left(\widetilde{Q}_h^{k'} - Q_h^{\pi_k} + \left[\widetilde{Q}_h^k - \widetilde{Q}_h^{k'}\right]_+\right)(x_h^k, a_h^k).$$

Applying Lemma A.2 with the $k' = k_{\tau_{\text{last}}(t)+1}$-th episode (so that there are $\tau_{\text{last}}(t)$ visitations to $(x_h^k, a_h^k)$ prior to the $k'$-th episode), we have the bound

$$\left(\widetilde{Q}_h^{k'} - Q_h^{\pi_k}\right)(x_h^k, a_h^k) \leq \left(\widetilde{Q}_h^{k'} - Q_h^\star\right)(x_h^k, a_h^k) + (Q_h^\star - Q_h^{\pi_k})(x_h^k, a_h^k)$$

$$\leq \alpha_{\tau_{\text{last}}(t)}^0 H + \sum_{i=1}^{\tau_{\text{last}}(t)} \alpha_{\tau_{\text{last}}(t)}^i \widetilde{\phi}_{h+1}^{k_i} + \beta_{\tau_{\text{last}}(t)} - \widetilde{\phi}_{h+1}^k + \widetilde{\delta}_{h+1}^k + \xi_{h+1}^k, \tag{3}$$

where we recall that $\beta_t = 2\sum_i \alpha_t^i b_i = \Theta(\sqrt{H^3\ell/t})$ and $\xi_{h+1}^k := [(\widehat{\mathbb{P}}_h^{k_i} - \mathbb{P}_h)(V_{h+1}^\star - V_{h+1}^{\pi_k}](x_h^k, a_h^k)$. By Lemma A.4, the perturbation term $[\widetilde{Q}_h^k - \widetilde{Q}_h^{k'}]_+$ can be bounded as

$$[\widetilde{Q}_h^k - \widetilde{Q}_h^{k'}]_+(x_h^k, a_h^k) \leq \beta_t + \sum_{i=\tau_{\text{last}}(t)+1}^{t} \alpha_t^i \widetilde{\phi}_{h+1}^{k_i} + C\sqrt{\frac{H^3\ell}{t}}. \tag{4}$$

We now study the effect of adding (4) onto (3). The term $C\sqrt{H^3\ell/t}$ in (4) and $\beta_{\tau_{\text{last}}(t)}$ in (3) can be both absorbed into $\beta_t$ (as $\beta_t \geq 2\sqrt{H^3\ell/t}$ and $\beta_{\tau_{\text{last}}(t)} \leq \sqrt{1+\eta}\beta_t$), so these together is bounded by $C'\beta_t$ where $C'$ is an absolute constant.

Adding (4) onto (3), we obtain

$$\widetilde{\delta}_h^k \leq \underbrace{\alpha_{\tau_{\text{last}}(t)}^0 H}_{\text{I}} + \underbrace{\sum_{i=1}^{\tau_{\text{last}}(t)} \alpha_{\tau_{\text{last}}(t)}^i \widetilde{\phi}_{h+1}^{k_i} + \sum_{i=\tau_{\text{last}}(t)+1}^{t} \alpha_t^i \widetilde{\phi}_{h+1}^{k_i} + C'\beta_t - \widetilde{\phi}_{h+1}^k + \widetilde{\delta}_{h+1}^k + \xi_{h+1}^k}_{\text{II}}.$$

We now sum the above bound over $k \in [K]$. For term I, it equals $H$ only when $\tau_{\text{last}}(t) = 0$, which happens only if $t = 0$, so the sum over $k$ is upper bounded by $SAH$.

For term II, we consider the coefficient in front of $\widetilde{\phi}_{h+1}^{k'}$ for each $k' \in [K]$ when summing over $k$. Let $n_h^k$ denote the number of visitations to $(x_h^k, a_h^k)$ prior to the $k$-th episode. For each $k'$, $\widetilde{\phi}_{h+1}^{k'}$ is counted if $i = n_h^{k'}$ and $(x_h^k, a_h^k) = (x_h^{k'}, a_h^{k'})$. We use $t$ to denote $n_h^k$, then an $\alpha_{\tau_{\text{last}}(t)}^{n_h^{k'}}$ appears if $\tau_{\text{last}}(t) \geq n_h^{k'}$, and an $\alpha_t^{n_h^{k'}}$ appears if $\tau_{\text{last}}(t) + 1 \leq n_h^{k'} \leq t$. So the total coefficient in front of $\widetilde{\phi}_{h+1}^{k'}$ is at most

$$\sum_{t: t \geq n_h^{k'}, \tau_{\text{last}}(t) \leq n_h^{k'}-1} \alpha_t^{n_h^{k'}} + \sum_{t: \tau_{\text{last}}(t) \geq n_h^{k'}} \alpha_{\tau_{\text{last}}(t)}^{n_h^{k'}},$$

for each $k' \in [K]$. Choosing $\eta = \frac{1}{2H(H+1)}$ and $r_\star = \left\lceil \frac{\log(10H^2)}{\log(1+\eta)} \right\rceil$, applying Lemma A.5, the above is upper bounded by $1 + 3/H$.

For the remaining terms, we can adapt the proof of Theorem 1 in [1] and obtain a propagation of error inequality, and deduce (as $(1 + 3/H)^H = O(1)$) that the regret is bounded by $O(\sqrt{H^4 SAT\ell})$. This concludes the proof.

**Part II: Bound on local switching cost**  For each $(h, x) \in [H] \times \mathcal{S}$ and each action $a \in \mathcal{A} = [A]$, either it is in stage I, which induces a switching cost of at most $\tau(r_\star)$, or it is in stage II, which according to the triggering sequence induces a switching cost of

$$\tau(r_\star) + r_a - r_\star \leq \tau(r_\star) + r_a,$$

where $r_a$ is the final index for action $a$ satisfying

$$\sum_{a=1}^A \lceil (1 + \eta)^{r_a} \rceil \leq K + H,$$

(define $r_a = 0$ if action $a$ has not reached the second stage.) Applying Jensen's inequality gives that

$$\sum_{a=1}^A r_a \leq \frac{A \log((K+H)/A)}{\log(1 + \eta)} = O\left(H^2 A \log(K/A)\right)$$

So the switching cost for $(h, x)$ can be bounded as

$$
\begin{aligned}
&A\tau(r_\star) + \sum_{a=1}^A r_a \\
&\leq A \lceil (1+\eta)^{r_\star} \rceil + O\left(H^2 A \log(K/A)\right) \leq A \lceil (1+\eta) \cdot 10H^2 \rceil + O\left(H^2 A \log(K/A)\right) \\
&\leq 20H^2 A + O\left(H^2 A \log(K/A)\right) = O\left(H^2 A \log(K/A)\right).
\end{aligned}
$$

Multiplying the above by $HS$ (the number of $(h, x)$ pairs) gives the desired bound. $\qquad\square$

# B  Q-learning with UCB2-Bernstein exploration

## B.1  Algorithm description

We present the algorithm, Q-Learning with UCB2-Bernstein (UCB2B) exploration, in Algorithm 1 below.

## B.2  Proof of Theorem 3

We first present the analogs of Lemmas that we used in the proof of Theorem 2.

**Algorithm 1** Q-learning with UCB2-Bernstein (UCB2B) Exploration

---

**input** Parameter $\eta \in (0,1)$, $r_\star \in \mathbb{Z}_{>0}$, and $c > 0$.

   **Initialize:** $\widetilde{Q}_h(x,a) \leftarrow H$, $Q_h \leftarrow \widetilde{Q}_h$, $N_h(x,a) \leftarrow 0$ for all $(x,a,h) \in \mathcal{S} \times \mathcal{A} \times [H]$.

   **for** episode $k = 1,\ldots,K$ **do**

      Receive $x_1$.

      **for** step $h = 1,\ldots,H$ **do**

         Take action $a_h \leftarrow \arg\max_{a'} Q_h(x_h, a')$, and observe $x_{h+1}$.

         $t = N_h(x_h, a_h) \leftarrow N_h(x_h, a_h) + 1$.

         $\mu_h(x_h, a_h) \leftarrow \mu_h(s_h, a_h) + \widetilde{V}_{h+1}(x_{h+1})$.

         $\sigma_h(x_h, a_h) \leftarrow \sigma_h(x_h, a_h) + \left(\widetilde{V}_{h+1}(x_{h+1})\right)^2$.

         $W_t(x_h, a_h, h) = \frac{1}{t}\left(\sigma_h(x_h, a_h) - (\mu_h(x_h, a_h))^2\right)$.

         $\beta_t(x_h, a_h, h) \leftarrow \min\left\{c_1\left(\sqrt{\frac{H}{t}(W_t(x_h,a_h,h)+H)\ell} + \frac{\sqrt{H^7SA\cdot\ell}}{t}\right), c_2\sqrt{\frac{H^3\ell}{t}}\right\}$.

         $b_t \leftarrow \frac{\beta_t(x_h,a_h,h)-(1-\alpha_t)\beta_{t-1}(x_h,a_h,h)}{2\alpha_t}$ (Bernstein-type bonus).

         $\widetilde{Q}_h(x_h, a_h) \leftarrow (1-\alpha_t)\widetilde{Q}_h(x_h, a_h) + \alpha_t[r_h(x_h, a_h) + \widetilde{V}_{h+1}(x_{h+1}) + b_t]$.

         $\widetilde{V}_h(x_h) \leftarrow \min\left\{H, \max_{a'\in\mathcal{A}}\widetilde{Q}_h(x_h, a')\right\}$.

         **if** $t \in \{t_n\}_{n\geq 1}$ (where $t_n$ is defined in (1)) **then**

            (Update policy) $Q_h(x_h, \cdot) \leftarrow \widetilde{Q}_h(x_h, \cdot)$.

         **end if**

      **end for**

   **end for**

---

**Lemma B.1** ($\widetilde{Q}$ is optimistic and accurate for the Bernstein case; Lemma C.1 & C.4, [1]). *We have for all* $(h, x, a, k) \in [H] \times \mathcal{S} \times \mathcal{A} \times [K]$ *that*

$$\widetilde{Q}_h^k(x,a) - Q_h^\star(x,a)$$
$$= \alpha_t^0(H - Q_h^\star(x,a)) +$$
$$\sum_{i=1}^{t} \alpha_t^i\left(r_h(x,a) + \widetilde{V}_{h+1}^{k_i}(x_{h+1}^{k_i}) - V_{h+1}^\star(x_{h+1}^{k_i}) + \left[\left(\widehat{\mathbb{P}}_h^{k_i} - \mathbb{P}_h\right)V_{h+1}^\star\right](x,a) + b_i\right),$$

*where* $[\widehat{\mathbb{P}}_h^{k_i} V_{h+1}](x,a) := V_{h+1}(x_{h+1}^{k_i})$.

*Further, with probability at least* $1 - p$, *under the choice of* $b_t$ *and* $\beta_t$ *in Algorithm 1, we have for all* $(h, x, a, k)$ *that*

$$0 \leq \widetilde{Q}_h^k(x,a) - Q_h^\star(x,a) \leq \alpha_t^0 H + \sum_{i=1}^{t} \alpha_t^i(\widetilde{V}_{h+1}^{k_i} - V_{h+1}^\star)(x_{h+1}^{k_i}) + \beta_t.$$

The following Lemma is the analog of Lemma A.3 in the Bernstein case.

**Lemma B.2** (Max error under delayed policy update). *We have*

$$\widetilde{\delta}_h^k \leq \left(\max\left\{\widetilde{Q}_h^{k'}, \widetilde{Q}_h^k\right\} - Q_h^{\pi_k}\right)(x_h^k, a_h^k) = \left(\widetilde{Q}_h^{k'} - Q_h^{\pi_k} + \left[\widetilde{Q}_h^k - \widetilde{Q}_h^{k'}\right]_+\right)(x_h^k, a_h^k).$$

*where* $k' = k_{\tau_{\mathrm{last}}(t)+1}$ *(which depends on k.) In particular, if* $t = \tau_{\mathrm{last}}(t)$, *then* $k = k'$ *and the upper bound reduces to* $(\widetilde{Q}_h^{k'} - Q_h^{\pi_k})(x_h^k, a_h^k)$.

The proof of Lemma B.2 can be adapted from the proof of Lemma A.3. The following Lemma is the analog of Lemma A.4 in the Bernstein case.

**Lemma B.3** (Perturbation bound on $(\widetilde{Q}_h^k - \widetilde{Q}_h^{k'})_+$). *For any* $k$ *such that* $k > k'$ *(so that the perturbation term is non-zero), we have*

$$\left[\widetilde{Q}_h^k - \widetilde{Q}_h^{k'}\right]_+(x_h^k, a_h^k) \leq \beta_t + \sum_{i=\tau_{\mathrm{last}}(t)+1}^{t} \alpha_t^i\widetilde{\phi}_{h+1}^{k_i} + \overline{\zeta}_h^k,$$

*where*

$$\overline{\zeta}_h^k := \left| \sum_{i=\tau_{\text{last}}(t)+1}^{t} \alpha_t^i [(\widehat{\mathbb{P}}_h^k - \mathbb{P}_h) V_{h+1}^\star](x_h^k, a_h^k) \right|.$$

The proof of Lemma B.3 can be adapted from the proof of Lemma A.4, but we used a finer bound on the summation $\overline{\zeta}_h^k$ over $k \in [K]$ in the proof of Theorem 3.

**Lemma B.4** (Variance is bounded and $W_t$ is accurate; Lemma C.5 & C.6, [1]). *There exists an absolute constant c, such that*

$$\sum_{k=1}^{K} \sum_{h=1}^{H} \mathbb{V}_h V_{h+1}^{\pi_k}(x_h^k, a_h^k) \le c(HT + H^3 \ell),$$

*w.p. at least $(1-p)$.*

*Further, w.p. at least $(1-4p)$, there exists an absolute constant $c > 0$ such that, letting $(x,a) = (x_h^k, a_h^k)$ and $t = n_h^k = N_h^k(x,a)$, we have*

$$W_t(x,a,h) \le \mathbb{V}_h V_{h+1}^{\pi_k}(x,a) + 2H(\widetilde{\delta}_{h+1}^k + \xi_{h+1}^k) + c\left( \frac{SA\sqrt{H^7 \ell}}{t} + \sqrt{\frac{SAH^7 \ell}{t}} \right)$$

*for all $(k,h) \in [K] \times [H]$, where the variance operator $\mathbb{V}_h$ is defined by*

$$[\mathbb{V}_h V_{h+1}](x,a) := Var_{x' \sim \mathbb{P}_h(\cdot|x,a)}(V_{h+1}(x')) = \mathbb{E}_{x' \sim \mathbb{P}_h(\cdot|x,a)} \left[ V_{h+1}(x') - [\mathbb{P}_h V_{h+1}](x,a) \right]^2.$$

Now, it is ready to present the proof of Theorem 3.

**Theorem 3** (Q-learning with UCB2B, restated). *Choosing $\eta = \frac{1}{2H(H+1)}$ and $r_\star = \left\lceil \frac{\log(10H^2)}{\log(1+\eta)} \right\rceil$, with probability at least $1-p$, the regret of Algorithm 1 is bounded by $O(\sqrt{H^3 SAT \ell^2} + \sqrt{S^3 A^3 H^9 \ell^4})$, where $\ell := \log(SAT/p)$ is a log factor. Further, the local switching cost is bounded as $N_{\text{switch}} \le O(H^3 SA \log(K/A))$.*

**Proof of Theorem 3**

By Lemma B.2, we have

$$\widetilde{\delta}_h^k \le \left( \widetilde{Q}_h^{k'} - Q_h^{\pi_k} + \left[ \widetilde{Q}_h^k - \widetilde{Q}_h^{k'} \right]_+ \right)(x_h^k, a_h^k).$$

Applying Lemma B.1 with the $k' = k_{\tau_{\text{last}}(t)+1}$-th episode (so that there are $\tau_{\text{last}}(t)$ visitations to $(x_h^k, a_h^k)$ prior to the $k'$-th episode), we have the bound

$$\left( \widetilde{Q}_h^{k'} - Q_h^{\pi_k} \right)(x_h^k, a_h^k) \le \left( \widetilde{Q}_h^{k'} - Q_h^\star \right)(x_h^k, a_h^k) + (Q_h^\star - Q_h^{\pi_k})(x_h^k, a_h^k)$$

$$\le \alpha_{\tau_{\text{last}}(t)}^0 H + \sum_{i=1}^{\tau_{\text{last}}(t)} \alpha_{\tau_{\text{last}}(t)}^i \widetilde{\phi}_{h+1}^{k_i} + \beta_{\tau_{\text{last}}(t)} - \widetilde{\phi}_{h+1}^k + \widetilde{\delta}_{h+1}^k + \xi_{h+1}^k. \tag{5}$$

By Lemma B.3, the perturbation term $[\widetilde{Q}_h^k - \widetilde{Q}_h^{k'}]_+$ can be bounded as

$$[\widetilde{Q}_h^k - \widetilde{Q}_h^{k'}]_+(x_h^k, a_h^k) \le \beta_t + \sum_{i=\tau_{\text{last}}(t)+1}^{t} \alpha_t^i \widetilde{\phi}_{h+1}^{k_i} + \overline{\zeta}_h^k. \tag{6}$$

Thus, adding (6) onto (5), we obtain

$$\widetilde{\delta}_h^k \le \underbrace{\alpha_{\tau_{\text{last}}(t)}^0 H}_{\text{I}} + \underbrace{\sum_{i=1}^{\tau_{\text{last}}(t)} \alpha_{\tau_{\text{last}}(t)}^i \widetilde{\phi}_{h+1}^{k_i} + \sum_{i=\tau_{\text{last}}(t)+1}^{t} \alpha_t^i \widetilde{\phi}_{h+1}^{k_i}}_{\text{II}} + \underbrace{\overline{\zeta}_h^k}_{\text{III}}$$

$$+ \underbrace{\beta_{\tau_{\text{last}}(t)}}_{\text{IV}} + \underbrace{\xi_{h+1}^k}_{\text{V}} - \widetilde{\phi}_{h+1}^k + \widetilde{\delta}_{h+1}^k + \beta_t.$$

We now sum the above bound over $k \in [K]$ and $h \in [H]$. For term I, it equals $H$ only when $\tau_{\text{last}}(t) = 0$, which happens only if $t = 0$, so the sum over $k$ is upper bounded by $SAH$.

For term II, we follow the same argument in the proof of Theorem 2 and obtain:

$$\sum_{k=1}^{K} \left( \sum_{i=1}^{\tau_{\text{last}}(t)} \alpha_{\tau_{\text{last}}(t)}^{i} \widetilde{\phi}_{h+1}^{k_i} + \sum_{i=\tau_{\text{last}}(t)+1}^{t} \alpha_t^i \widetilde{\phi}_{h+1}^{k_i} \right) \leq \left( 1 + \frac{3}{H} \right) \sum_{k=1}^{K} \widetilde{\phi}_{h+1}^{k}$$

For term III, we first apply the Azuma-Hoeffding inequality to get that

$$\bar{\zeta}_h^k \leq c \sqrt{\sum_{i=\tau_{\text{last}}(t)+1}^{t} (\alpha_t^i)^2 H^2 \ell}$$

uniformly over $(h, k)$, then we sum it the above over $k \in [K]$, and then we obtain

$$\sum_{k=1}^{K} \bar{\zeta}_h^k \leq cH\sqrt{\ell} \sum_{k=1}^{K} \sqrt{\sum_{i=\tau_{\text{last}}(t)+1}^{t} (\alpha_t^i)^2} \leq cH\sqrt{\ell} \sum_{k=1}^{K} \sqrt{\sum_{i=\left\lceil \frac{n_h^k}{1+\eta} \right\rceil}^{n_h^k} \left( \alpha_{n_h^k}^i \right)^2}$$

$$\leq cH\sqrt{\ell} \sum_{k=1}^{K} \sqrt{\left( n_h^k - \left\lceil \frac{n_h^k}{1+\eta} \right\rceil \right) \left( \max_{i \in [n_h^k]} \alpha_{n_h^k}^i \right)^2}$$

$$\leq cH\sqrt{\ell} \sum_{k=1}^{K} \sqrt{\eta n_h^k \frac{4H^2}{(n_h^k)^2}}$$

$$\leq cH\sqrt{\ell} \sum_{k=1}^{K} \sqrt{\frac{1}{n_h^k}} \overset{(i)}{=} cH\sqrt{\ell} \sum_{x,a} \sum_{n=1}^{N_h^k(s,a)} \sqrt{\frac{1}{n}} \overset{(ii)}{\leq} cH\sqrt{SAK\ell}, \tag{7}$$

where (i) follows the fact $\sum_{s,a} N_h^K(x,a) = K$, and (ii) follows the property that the LHS of (ii) is maximized when $N_h^K(x,a) = K/SA$ for all $x, a$.

For term IV, we have

$$\sum_{k=1}^{K} \sum_{h=1}^{H} \beta_{\tau_{\text{last}}(n_h^k)} \leq c_1 \sum_{k=1}^{K} \sum_{h=1}^{H} \left( \sqrt{\frac{H}{\tau_{\text{last}}(n_h^k)} (W_{\tau_{\text{last}}(n_h^k)}(x,a,h) + H)\ell} + \frac{\sqrt{H^7 SA} \cdot \ell}{\tau_{\text{last}}(n_h^k)} \right) \tag{8}$$

by our choice of $\beta_t$ in Algorithm 1. We first upper bound summation the $W_{\tau_{\text{last}}(n_h^k)}(x,a,h)$ term as follows

$$\sum_{k=1}^{K} \sum_{h=1}^{H} W_{\tau_{\text{last}}(n_h^k)}(x,a,h)$$

$$\overset{(i)}{\leq} \sum_{k=1}^{K} \sum_{h=1}^{H} \left[ \mathbb{V}_h V_{h+1}^{\pi_k}(x,a) + 2H(\delta_{h+1}^k + \xi_{h+1}^k) + c \left( \frac{SA\sqrt{H^7\ell}}{\tau_{\text{last}}(n_h^k)} + \sqrt{\frac{SAH^7\ell}{\tau_{\text{last}}(n_h^k)}} \right) \right]$$

$$\overset{(ii)}{\leq} \sum_{k=1}^{K} \sum_{h=1}^{H} \left[ \mathbb{V}_h V_{h+1}^{\pi_k}(x,a) + 2H(\delta_{h+1}^k + \xi_{h+1}^k) + c(1+\eta) \left( \frac{SA\sqrt{H^7\ell}}{n_h^k} + \sqrt{\frac{SAH^7\ell}{n_h^k}} \right) \right]$$

$$\overset{(iii)}{\leq} \sum_{k=1}^{K} \sum_{h=1}^{H} \left[ \mathbb{V}_h V_{h+1}^{\pi_k}(x,a) + 2H(\delta_{h+1}^k + \xi_{h+1}^k) \right] + c(1+\eta) \left( S^2 A^2 \sqrt{H^9 \ell^3} + SA\sqrt{H^8 T\ell} \right)$$

$$\overset{(iv)}{\leq} 2H \sum_{k=1}^{K} \sum_{h=1}^{H} (\delta_{h+1}^k + \xi_{h+1}^k) + c' \left( HT + H^3\ell + S^2 A^2 \sqrt{H^9 \ell^3} + SA\sqrt{H^8 T\ell} \right), \tag{9}$$

where inequalities (i) and (iv) follow from Lemma B.4, inequality (ii) follows from $\tau_{\text{last}}(n_h^k) \geq n_h^k/(1+\eta)$, and inequality (iii) uses the properties that $\sum_{k=1}^{K} (n_h^k)^{-1}$ and $\sum_{k=1}^{K} (n_h^k)^{-1/2}$ are maximized when $N_h^K(x,a) = K/SA$ for all $x, a$ (similar to (7)).

We now consider the first term in (9). By the Azuma-Hoeffding inequality, we have

$$
\left| \sum_{h'=h}^{H} \sum_{k=1}^{K} \xi_{h'+1}^{k} \right| \leq \left| \sum_{h'=h}^{H} \sum_{k=1}^{K} [(\widehat{\mathbb{P}}_{h'}^{k_i} - \mathbb{P}_h)(V_{h'+1}^{\star} - V_{h'+1}^{\pi_k})](x_{h'}^k, a_{h'}^k) \right| \leq O(H\sqrt{T\ell}), \quad (10)
$$

w.p. $1 - p$ for all $h \in [H]$. Recall $\beta_t(x, a, h) \leq c\sqrt{H^3\ell/t}$, we can simply obtain

$$
\sum_{k=1}^{K} \delta_h^k \leq O(\sqrt{H^4 SAT\ell}), \quad (11)
$$

for all $h \in [H]$ by adapting the proof of Theorem 2. Then, using (10) and (11), we obtain the upper bound of the summation of $W_{\tau_{\text{last}}(n_h^k)}(x, a, h)$ term for $h \in [H]$ and $k \in [K]$

$$
\sum_{k=1}^{K} \sum_{h=1}^{H} W_{\tau_{\text{last}}(n_h^k)}(x, a, h)
$$
$$
2H \sum_{k=1}^{K} \sum_{h=1}^{H} (\delta_{h+1}^k + \xi_{h+1}^k) + c' \left( HT + H^3\ell + S^2 A^2 \sqrt{H^9\ell^3} + SA\sqrt{H^8 T\ell} \right)
$$
$$
\leq O \left( HT + S^2 A^2 H^7 \ell + S^2 A^2 \sqrt{H^9\ell^3} \right). \quad (12)
$$

Now it is ready to upper bounded the summation of the first term in (8),

$$
\sum_{k=1}^{K} \sum_{h=1}^{H} \sqrt{\frac{H}{\tau_{\text{last}}(n_h^k)} (W_{\tau_{\text{last}}(n_h^k)}(x, a, h) + H)\ell}
$$
$$
\overset{(i)}{\leq} \sqrt{\left( \sum_{k=1}^{K} \sum_{h=1}^{H} (W_{\tau_{\text{last}}(n_h^k)}(x, a, h) + H) \right) \left( \sum_{k=1}^{K} \sum_{h=1}^{H} \frac{H}{\tau_{\text{last}}(n_h^k)} \right) \ell}
$$
$$
\overset{(ii)}{\leq} (1+\eta) \sqrt{\sum_{k=1}^{K} \sum_{h=1}^{H} W_{\tau_{\text{last}}(n_h^k)}(x, a, h)} \cdot \sqrt{H^2 SA\ell^2} + (1+\eta)\sqrt{H^3 SAT\ell^2}
$$
$$
\overset{(iii)}{\leq} O(\sqrt{H^3 SAT\ell^2}) \quad (13)
$$

where inequality (i) follows from the Cauchy–Schwarz inequality, inequality (ii) follows from the facts that $\tau_{\text{last}}(n_h^k) \geq n_h^k/(1+\eta)$ and $\sum_{k=1}^{K} (n_h^k)^{-1}$ is maximized when $N_h^K(x, a) = K/SA$ for all $x, a$, and inequality (iii) follows from (12).

The summation of the second term in (8) can be upper bounded by

$$
\sum_{k=1}^{K} \sum_{h=1}^{H} \frac{\sqrt{H^7 SA} \cdot \ell}{\tau_{\text{last}}(n_h^k)} \leq \sum_{k=1}^{K} \sum_{h=1}^{H} \frac{(1+\eta)\sqrt{H^7 SA} \cdot \ell}{n_h^k} \leq (1+\eta)\sqrt{H^9 S^3 A^3 \ell^4}, \quad (14)
$$

by following $\tau_{\text{last}}(n_h^k) \geq n_h^k/(1+\eta)$ and $1 + 1/2 + 1/3 + \cdots \leq \ell$.

Putting (8), (13), and (14) together, we have

$$
\sum_{k=1}^{K} \sum_{h=1}^{H} \beta_{\tau_{\text{last}}(n_h^k)} \leq O \left( \sqrt{H^3 SAT\ell^2} + \sqrt{S^3 A^3 H^9 \ell^4} \right).
$$

For the remaining terms, we can adapt the proof of Theorem 2 in [1] and obtain a propagation of error inequality. Thus, we deduce that the regret is bounded by $O(\sqrt{H^3 SAT\ell^2} + \sqrt{S^3 A^3 H^9 \ell^4})$. The bound on local switching cost can be adapted from the proof of Theorem 2. This concludes the proof. □

# C  Proof of Corollary 4

Consider first Q-Learning with UCB2H exploration. By Theorem 2, we know that the regret is bounded by $\widetilde{O}(\sqrt{H^4SAT})$ with high probability, that is, we have

$$\sum_{k=1}^{K} V_1^\star(x_1) - V_1^{\pi_k}(x_1) \le \widetilde{O}(\sqrt{H^4SAT}).$$

Now, define a stochastic policy $\widehat{\pi}$ as

$$\widehat{\pi} = \frac{1}{K}\sum_{k=1}^{K}\pi_k.$$

By definition we have

$$\mathbb{E}\left[V_1^\star(x_1) - V_1^{\widehat{\pi}}(x_1)\right] = \frac{1}{K}\sum_{k=1}^{K}[V_1^\star(x_1) - V_1^{\pi_k}(x_1)] \le \widetilde{O}\left(\frac{\sqrt{H^4SAT}}{K}\right) = \widetilde{O}\left(\sqrt{\frac{H^5SA}{K}}\right).$$

So by the Markov inequality, we have with high probability that

$$V_1^\star(x_1) - V_1^{\widehat{\pi}}(x_1) \le \widetilde{O}\left(\sqrt{\frac{H^5SA}{K}}\right).$$

Taking $K = \widetilde{O}(H^5SA/\varepsilon^2)$ bounds the above by $\varepsilon$.

For Q-Learning with UCB2B exploration, the regret bound is $\widetilde{O}(\sqrt{H^3SAT})$. A similar argument as above gives that $K = \widetilde{O}(H^4SA/\varepsilon^2)$ episodes guarantees an $\varepsilon$ near-optimal policy with high probability. $\qquad\square$

# D  Proof of Theorem 5

We first present the concurrent version of low-switching cost Q-learning with {UCB2H, UCB2B} exploration.

**Algorithm description**  At a high level, our algorithm is a very intuitive parallelization of the vanilla version – we "parallelize as much as you can" until we have to switch.

More concretely, suppose the policy $Q_h$ has been switched $(t-1)$ times and we have a new policy yet to be executed. We execute this policy on all $M$ machines, and read the observed trajectories from machine 1 to $M$ to determine a number $m \in \{1,\dots,M\}$ such that the policy needs to be switched (according to the UCB2 schedule) after $m$ episodes. We then only keep the data on machines $1,\dots,m$, use them to compute the next policy, and throw away all the rest of the data on machines $m+1,\dots,M$. The full algorithm is presented in Algorithm 2.

## D.1  Proof of Theorem 5

The way that Algorithm 2 is constructed guarantees that its execution path is *exactly equivalent* (i.e. equal in distribution) to the execution path of the vanilla non-parallel Q-Learning with UCB2{H, B} exploration, except that it does not fully utilize the data on all $M$ machines and needs to throw away some data. As a corollary, if the non-parallel version plays $L_t$ episodes in between the $(t-1)$-th and $t$-th switch, then the parallel/concurrent version will play the same episodes in $\lceil L_t/M\rceil$ rounds.

Now, suppose we wish to play a total of $K$ episodes concurrently with $M$ machines, and the corresponding non-parallel version of Q-learning is guaranteed to have at most $N_{\text{switch}}$ local switches with $L_t$ episodes played before each switch. Let $R$ denote the total number of rounds, then we have

$$R = \sum_{t=1}^{N_{\text{switch}}} r_t = \sum_{t=1}^{N_{\text{switch}}} \left\lceil \frac{L_t}{M} \right\rceil \le \sum_{t=1}^{N_{\text{switch}}} \left(1 + \frac{L_t}{M}\right) \le N_{\text{switch}} + \frac{K}{M}.$$

---
**Algorithm 2** Concurrent Q-learning with UCB2 scheduling
---
**input** One of the UCB2-{Hoeffding, Bernstein} bonuses for updating $\widetilde{Q}$.
   **Initialize:** $\widetilde{Q}_h(x,a) \leftarrow H, Q_h \leftarrow \widetilde{Q}_h, t \leftarrow 1$.
   **while** stopping criterion not satisfied **do**
      **for** rounds $r_t = 1, 2, \ldots$ **do**
         Play according to $Q_h$ concurrently on all $M$ machines and store the trajectories.
         Aggregate the trajectories and feed them sequentially into the UCB2 scheduling to determine whether a switch is needed.
         **if** Switch is needed after $m \in \{1, \ldots, M\}$ episodes **then**
            **BREAK**
         **end if**
      **end for**
      Update the policy $\widetilde{Q}_h$ from all the $M(r_t-1)+m$ stored trajectories using {Hoeffding, Bernstein} bonus.
      Set $Q_h(\cdot,\cdot) \leftarrow \widetilde{Q}_h(\cdot,\cdot)$ and $t \leftarrow t + 1$.
   **end while**
---

Now, to find $\varepsilon$ near-optimal policy, we know by Corollary 4 that Q-learning with {UCB2H, UCB2B} exploration requires at most

$$K = O\left(\frac{H^{\{5,4\}}SA\log(HSA)}{\varepsilon^2}\right)$$

episodes. Further, choosing $K$ as above, by Theorem 2 and 3, the switching cost is bounded as

$$N_{\text{switch}} \le O\left(H^3 SA \log(K/A)\right) = O\left(H^3 SA \log(HSA/\varepsilon)\right).$$

Plugging these into the preceding bound on $R$ yields

$$R \le O\left(H^3 SA \log(HSA/\varepsilon) + \frac{H^{\{5,4\}}SA\log(HSA)}{\varepsilon^2 M}\right) = \widetilde{O}\left(H^3 SA + \frac{H^{\{5,4\}}SA}{\varepsilon^2 M}\right),$$

the desired result. $\qquad\square$

### D.2 Concurrent algorithm with mistake bound

Our concurrent algorithm (Algorithm 2 can be converted straightforwardly to an algorithm with low mistake bound. Indeed, for any given $\varepsilon$, by Theorem 5, we obtain an $\varepsilon$ near-optimal policy with high probability by running Algorithm 2 for

$$\widetilde{O}\left(H^3 SA + \frac{H^{\{5,4\}}SA}{\varepsilon^2 M}\right)$$

rounds. We then run this $\varepsilon$ near-optimal policy forever and are guaranteed to make no mistake.

For such an algorithm, with high probability, "mistakes" can only happen in the exploration phase. Therefore the total amount of "mistakes" (performing an $\varepsilon$ sub-optimal action) is upper bounded by the above number of exploration rounds multiplied by $HM$, as each round consists of at most $M$ machines[1] each performing $H$ actions. This yields a mistake bound

$$\widetilde{O}\left(H^4 SAM + \frac{H^{\{6,5\}}SA}{\varepsilon^2}\right)$$

as desired.

## E   Proof of Theorem 6

Recall that $\mathcal{M}$ denotes the set of all MDPs with horizon $H$, state space $S$, action space $A$, and deterministic rewards in $[0, 1]$. Let $K$ be the number of episodes that we can run, and $\mathcal{A}$ be any RL

algorithm satisfying that

$$N_{\text{switch}} = \sum_{(h,x)} n_{\text{switch}}(h,x) \leq HSA/2$$

almost surely. We want to show that

$$\sup_{M \in \mathcal{M}} \mathbb{E}_{x_1,M} \left[ \sum_{k=1}^{K} V_1^{\star}(x_1) - V_1^{\pi_k}(x_1) \right] \geq \Omega(K),$$

i.e. the worst case regret is linear in $K$.

## E.1 Construction of prior

Let $a^{\star} : [H] \times [S] \to [A]$ denote a mapping that maps each $(h,x)$ to an action $a^{\star}(h,x) \in [A]$. There are $A^{HS}$ such mappings. For each $a^{\star}$, define an MDP $M_{a^{\star}}$ where the transition is uniform, i.e.

$$x_1 \sim \text{Unif}([S]), \quad x_{h+1}|x_h = x, a_h = a \sim \text{Unif}([S]) \quad \text{for all } (x,a) \in [S] \times [A], \ h \in [H]$$

and the reward is 1 if $a_h = a^{\star}(h,x_h)$ and 0 otherwise, that is,

$$r_h(x,a) = \mathbf{1} \left\{ a = a^{\star}(h,x) \right\}.$$

Essentially, $M_{a^{\star}}$ is just a $H$-fold connection of $S$ parallel bandits that are $A$-armed, where $a^{\star}(h,x)$ is the only optimal action at each $(h,x)$.

For such MDPs, as the transition does not depend on the policy, the value functions can be expressed explicitly as

$$\mathbb{E}_{x_1}[V_1^{\pi}(x_1)] = \frac{1}{S} \sum_{(h,x) \in [H] \times [S]} \mathbf{1} \left\{ \pi_h(x) = a^{\star}(h,x) \right\},$$

and we clearly have

$$\mathbb{E}_{x_1}[V_1^{\star}(x_1)] \equiv H.$$

## E.2 Minimax lower bound

Using the sup to average reduction with the above prior, we have the bound

$$\sup_{M \in \mathcal{M}} \mathbb{E}_{x_1,M} \left[ \sum_{k=1}^{K} V_1^{\star}(x_1) - V_1^{\pi_k}(x_1) \right] \geq \mathbb{E}_{a^{\star}} \mathbb{E}_{M_{a^{\star}}} \left[ KH - \sum_{k=1}^{K} V_1^{\pi_k}(x_1) \right]$$

$$= KH - \sum_{k=1}^{K} \mathbb{E}_{a^{\star},M_{a^{\star}}}[V_1^{\pi_k}(x_1)].$$

It remains to upper bound $\mathbb{E}_{a^{\star},M_{a^{\star}}}[V_1^{\pi_k}(x_1)]$ for each $k$.

For all $k \geq 1$, let

$$n_{\text{switch}}^{k}(h,x) := \sum_{j=1}^{k-1} \mathbf{1} \left\{ \pi_j^h(x) \neq \pi_{j+1}^h(x) \right\} \quad \text{and} \quad N_{\text{switch}}^{k} = \sum_{h,x} n_{\text{switch}}^{k}(h,x)$$

denote respectively the switching cost at a single $(h,x)$ and the total (local) switching cost. We use the switching cost to upper bound $\mathbb{E}_{a^{\star},M_{a^{\star}}}[V_1^{\pi_k}]$.

Let

$$A_k(h,x) := \left\{ \pi_1^h(x), \ldots, \pi_k^h(x) \right\} \subseteq [A]$$

denote the set of visited actions at timestep $h$ and state $x$. Observe that

$$\mathbb{E}_{a^{\star},M_{a^{\star}}}[V_1^{\pi_k}] = \frac{1}{S} \sum_{h,x} \mathbb{E} \left[ \mathbf{1} \left\{ a^{\star}(h,x) = \pi_k^h(x) \right\} \right] \leq \frac{1}{S} \sum_{h,x} \underbrace{\mathbb{E}[\mathbf{1} \left\{ a^{\star}(h,x) \in A_k(h,x) \right\}]}_{:= \Phi_k(h,x)}.$$

Therefore it suffices to bound $\Phi_k(h,x)$.

It is clear that algorithms that only switch to unseen actions can maximize the value function, so we henceforth restrict attention on these algorithms. Let $a^\star = a^\star(h, x)$ and $n_{\text{switch}}^k = n_{\text{switch}}^k(h, x)$ for convenience. Let

$$A_k(h, x) = \left\{a^1, a^2, \ldots, a^{n_{\text{switch}}^k + 1}\right\}$$

be the ordered set of unique actions that have been taken at $(h, x)$ throughout the execution of the algorithm. We have

$$\Phi_k(h, x) = \mathbb{P}(a^\star \in A_k(h, x)) = \mathbb{P}\left(\bigcup_{j \geq 1} \left\{n_{\text{switch}}^k + 1 \geq j, a^\star \notin \left\{a^1, a^2, \ldots, a^{j-1}\right\}, a^\star = a^j\right\}\right)$$

$$= \sum_{j \geq 1} \mathbb{P}(n_{\text{switch}}^k + 1 \geq j) \cdot \mathbb{P}(a^\star \notin \left\{a^1, a^2, \ldots, a^{j-1}\right\}, a^\star = a^j \mid n_{\text{switch}}^k + 1 \geq j).$$

Now, suppose we know that $n_{\text{switch}}^k + 1 \geq j$, then the algorithm have seen the reward on $a^1, \ldots, a^{j-1}$. By the uniform prior of $a^\star$, if the algorithm has observed the rewards for all $a \in S$ and found that $a^\star \notin S$, the corresponding posterior for $a^\star$ would be uniform on $[A] \setminus S$. Therefore, we have recursively that

$$\mathbb{P}(a^\star \notin \left\{a^1, \ldots, a^{j-1}\right\}, a^\star = a^j \mid n_{\text{switch}}^k + 1 \geq j) = \prod_{\ell=1}^{j-1} \frac{A - \ell}{A - \ell + 1} \cdot \frac{1}{A - j + 1} = \frac{1}{A}.$$

Substituting this into the preceding bound gives

$$\Phi_k(h, x) = \frac{1}{A} \sum_{j \geq 1} \mathbb{P}(n_{\text{switch}}^k + 1 \geq j) = \frac{\mathbb{E}[n_{\text{switch}}^k + 1]}{A}$$

and thus

$$\mathbb{E}_{a^\star, M_{a^\star}}[V_1^{\pi_k}] \leq \frac{1}{S} \sum_{h,x} \Phi_k(h, x) \leq \frac{1}{S} \sum_{h,x} \frac{\mathbb{E}[n_{\text{switch}}^k(h, x) + 1]}{A} \leq \frac{H}{A} + \frac{\mathbb{E}[N_{\text{switch}}^k]}{SA}$$

As $N_{\text{switch}}^k \leq N_{\text{switch}}^K \leq HSA/2$ almost surely, we have for all $k$ that

$$\mathbb{E}_{a^\star, M_{a^\star}}[V_1^{\pi_k}] \leq H/A + H/2 \leq 3H/4$$

when $A \geq 4$ and thus the regret can be lower bounded as

$$KH - \sum_{k=1}^{K} \mathbb{E}_{a^\star, M_{a^\star}}[V_1^{\pi_k}] \geq KH/4,$$

concluding the proof. $\qquad\square$

## Footnotes

[1]To have a fair comparison with CMBIE, if a round does not utilize all $M$ machines, we still let all $M$ machines run and count their actions as their "mistakes".

## References

[1] C. Jin, Z. Allen-Zhu, S. Bubeck, and M. I. Jordan. Is Q-learning provably efficient? In *Advances in Neural Information Processing Systems*, pages 4868–4878, 2018.