[Reviews · NeurIPS 2019]

Reviewer 1



Originality: The notion of the "local switching cost" is novel. The algorithm for obtaining low-switching cost and low regret is rather straightforward -- a delayed update to the policy will do. The proof is also similar to that of [15]. The lower bound is simple but the message is clear. Quality & Clarity: The paper is well-written and clearly delivers the message. Significance: The notion of "local switching cost" makes a lot of sense in many real applications. Although the algorithm is rather straightforward, it also shows that a low switching cost algorithm can be achieved by a simple modification to some other low-regret algorithms. (In fact, I conjecture that every low-regret algorithm can be converted to a lower switching cost algorithm.) Line 36: "closed related" ==> "closely related"

Reviewer 2



i)Originality : Introduction of the notion of “local switching cost” which extends the notion switching cost in online learning to RL. They also present (two flavours of) a Q-learning algorithm that achieve the regret matching the previous work however with the added benefit of having lower local switching cost. ii)Quality : The motivation behind considering the problem of RL with low switching cost is clearly explained. They provide a lower bound on the switching cost of any algorithm with sublienar regret. The proposed algorithm follows a conservative approach to change policies, as is the requirement for low switching cost, however their analysis manages to recover the same regret bound as the previous work. They also show applications of their algorithms in concurrent RL. iii)Clarity: The paper is well written. Both the text and math are clear except few minor issues: a. Line 128: “...,so that the agent needs only play” b. Line 149: “Our algorithm maintains wo sets…” b. Line 254: “…,and it is a priori possible whether such a mismatch will blow up the propagation of error.” b. line 274: “.., at most off by a factor of O(H^2 log K)” iv)Significance : This paper addresses an important problem in RL which has strong practical motivations. They provide algorithms which are provably better at the considered problem than the previous work.

Reviewer 3



1. Technically, this paper is interesting but I just fail to understand the practical relevance of the problem. And since it puts together a bunch of well-understood ideas, it seems rather incremental. 2. Switching cost in online learning was introduced in the late 1980s but in a general way as opposed to a very specific cost function here. Furthermore, finite horizon MDPs are considered (as opposed to infinite horizon) which kind of negates the argument that it can be costly to switch policies. This is because the optimal policies in the finite horizon are non-stationary anyway. So, it may hardly matter that these change due to exploration. In my opinion, that problem will be much more interesting in a infinite horizon setting and with the both the local and the global switching cost functions. 3. In Theorem 2, it would be better to present exact bound on regret (if it is known) instead of just in O(.). I presume that from their analysis, the authors are able to achieve it. 4. I am not able to appreciate the application to Concurrent setting. Perhaps the authors can elaborate. We already know asynchronous QL converges under suitable recurrence conditions. So what is new here. 5. Why is LB in Section 7 interesting since it holds only when the switching cost is constrained.

[Author Response · NeurIPS 2019]

**Reviewer #3 & #4** We thank the reviewers for the positive feedback and helpful comments.

**Reviewer #5** We thank the reviewer for raising the questions, which we respond to below.

(2) "Why does switching cost make sense in finite-horizon MDPs?"

We believe there is some misunderstanding in the switching cost considered in this paper. First, our switching cost is *not* to measure the change of per-step policy from step to step, but rather the change of the **overall policies from episode to episode**. Concretely, what we meant by a policy $\pi$ is the aggregation of per-step policies:

$$\pi = \left\{ \pi^h : \mathcal{S} \to \mathcal{A} \right\}_{h=1}^{H}, \text{ or equivalently, a stationary mapping from } \mathcal{S} \times [H] \text{ to } \mathcal{A}$$

and our switching cost measures how $\pi_k$ differs from $\pi_{k+1}$, where $\pi_k$ is the policy deployed in episode $k$.

If we understand correctly, part of the reviewer's concern is that since optimal policies in finite-horizon MDPs are nonstationary, *to execute the policy one always needs to switch (between time steps)*, so switching is ubiquitous and should not be costly. **This argument confuses the two kinds of switching (between steps vs. between episodes)**: Switching between steps is cheap as it can be eliminated if we specify a full policy that takes the time step as part of its input. In contrast, switching between episodes can only be eliminated if our (augmented) policy takes data from all previous episodes as input, which is much more costly and sometimes impossible. See our response about motivation below for more examples where frequent policy switching between episodes is impractical.

Finally, we agree with the reviewer that switching cost can also be meaningfully studied in infinite-horizon MDPs, but that is asking the same question under a different MDP formulation, and does not differ from the question we consider here in a fundamentally different manner.

(1) "Practical relevance & technical contributions."

*Practical Motivation.* There are strong practical motivations for designing RL algorithms with low switching cost: it happens whenever changing the policy has a higher cost than using the same policy to gather data. Our introduction lists a variety of such scenarios in practice, but we'd like to make it clearer here by the following concrete example.

In personalized recommendation systems such as video recommendation on YouTube, the policy specifies what videos we recommend to users given their features. Standard (provably efficient) RL algorithms typically require adjusting its policy based on instantaneous feedback, so for example it needs to update the policy for User 2 after obtaining the feedback on User 1. But this is computationally impractical as there are so many users visiting at every second. In contrast, it makes more sense to use the same policy to aggregate data in a certain period before coming up with an improved policy, which is precisely the setting of low switching cost algorithms. We will improve the presentation of our motivation part to make this clearer to the audience.

*Technical Contributions.* Our key technical contribution is the establishment of finite-sample regret bound under delayed Q updates. As our algorithm plays greedily according to a delayed version of the Q estimate, a priori it may be the case that the errors caused by the delay may blow up and break the regret bound. We provide a tight control of the errors under UCB2 scheduling and show that these errors do not affect the regret bound. Our techniques are novel and we believe of broader interest for understanding the effect of delayed updates in exploration problems.

(4) "What is new about concurrent RL / relationship with asynchronous Q-Learning."

The concerns of concurrent RL and asynchronous Q-Learning are quite different: concurrent RL cares about the improvement in *speedup of exploration* when multiple machines can each play a copy of the MDP at the same time, whereas asynchronous Q-Learning is about the convergence under asynchrony of updating $Q(s, a)$ for different $(s, a)$. Indeed, existing results on asynchronous Q-Learning [e.g., Even-Dar and Mansour, 2003] only show convergence and do not provide an explicit sample complexity bound, and thus do not cover our result. Furthermore, classical Q-Learning analyses (including the asynchronous ones) dodge the challenge of exploration by assuming that all states are visited sufficiently often, but exploration is a key concern in our setting, so the results are generally incomparable. In fact, concurrent PAC RL is first studied by Guo and Brunskill in 2015, and is a relatively new research direction.

(5) "Why constrained setting in the lower bound."

We chose this simplified setting (lower bounding the regret for algorithms with switching cost $\leq HSA/2$) as the problem cannot be formulated in a standard fashion and reduced to information-theoretic tools. Indeed, as we assumed deterministic rewards, it is in principle possible for an algorithm with $HSA$ switches to achieve optimal regret. (Think about a bandit with $A$ arms.) We believe our lower bound provides a useful initial step in understanding the limit of switching costs; stronger lower bounds could be possible when rewards are stochastic, which we leave as future work.

# References

Eyal Even-Dar and Yishay Mansour. Learning rates for q-learning. *Journal of machine learning Research*, 5(Dec): 1–25, 2003.


[Meta-Review · NeurIPS 2019]

On balance, the initial reviews for this paper were positive, with one slightly negative review. In discussion it was felt that the the authors did a reasonable job of addressing the concerns of the reviewers, though there was still some concern that the result may not be "surprising". I encourage the authors to incorporate their responses to the reviewers into any future version of the paper.